# ComBack: A Versatile Dataset for Enhancing Compiler Backend Development Efficiency

**Ming Zhong**[1,2], **Fang Lyu**[1],* **Lulin Wang**[1], **Hongna Geng**[1,2], **Lei Qiu**[1,2]
**Huimin Cui**[1,2*], **Xiaobing Feng**[1,2]

[1] SKLP, Institute of Computing Technology, CAS, Beijing, China    [2] UCAS, Beijing, China

{zhongming21s, flv, wanglulin, genghongna, qiulei21b, cuihm, fxb}@ict.ac.cn

## Abstract

Compiler backends are tasked with generating executable machine code for processors. With the proliferation of diverse processors, it is imperative for programmers to tailor specific compiler backends to accommodate each one. Meanwhile, compiler backend development is a laborious and time-consuming task, lacking effective automation methods. Although language models have demonstrated strong abilities in code related tasks, the lack of appropriate datasets for compiler backend development limits the application of language models in this field.

In this paper, we introduce ComBack, the first public dataset designed for improving compiler backend development capabilities of language models. ComBack includes 178 backends for mainstream compilers and three tasks including statement-level completion, next-statement suggestion and code generation, representing common development scenarios. We conducted experiments by fine-tuning six pre-trained language models with ComBack, demonstrating its effectiveness in enhancing model accuracy across the three tasks. We further evaluated the top-performing model (CodeT5+) across the three tasks for new targets, comparing its accuracy with conventional methods (Fork-Flow), ChatGPT-3.5-Turbo, and Code-LLaMA-34B-Instruct. Remarkably, fine-tuned CodeT5+ with only 220M parameters on ComBack outperformed Fork-Flow methods significantly and surpassed ChatGPT and Code-LLaMA, suggesting potential efficiency improvements in compiler development. ComBack is avaliable at https://huggingface.co/datasets/docz1105/ComBack.

## 1 Introduction

A compiler is a fundamental computer software which translates source code from high-level programming language into low-level machine code, *e.g.*, assembly code, for target machines (referred to as "**target**" for simplicity).

As shown in Fig. 1, mainstream compilers like GCC [18] and LLVM [30] are divided into three parts: frontend, middle-end and backend. Specifically, the frontend related to programming languages, while the middle-end comprises of target-independent optimizations, and backend converts intermediate representation produced by the middle-end into machine code for various targets. The flourishing development of new processors nowadays demands continuous development for backends.

Compiler backend development necessitates a profound understanding of the target characteristics and the compiler infrastructure [19]. Thus, it entails prolonged development cycles and substantial manual

---

*Corresponding Author.

38th Conference on Neural Information Processing Systems (NeurIPS 2024) Track on Datasets and Benchmarks.

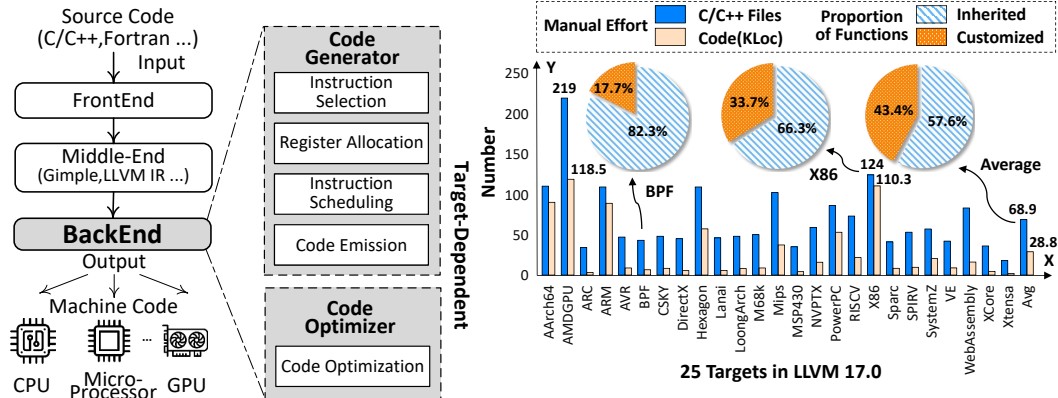

Figure 1: Compiler backend structure.

Figure 2: Heavy manual efforts in backends development for 25 targets in LLVM 17.0.1.

efforts. Data depicted in Fig. 2 underscores the magnitude of manual efforts and the distribution of functions across the development of compiler backends for 25 targets in LLVM 17.0.1 (latest released version). For instance, AMDGPU comprises 219 C++/C files, totaling 118.5 KLoC (Line of Code), while X86 comprises 124 files with 110.3 KLoC. On average, a LLVM backend in LLVM 17.0.1 consists of 68.9 files, encompassing 28.8 KLoC, indicating considerable manual efforts.

The emergence of AI has spurred considerable interest in leveraging its techniques for code-related tasks, such as code completion and generation [34, 14, 22, 50, 49, 21, 23, 45, 56]. Models like Code-LLaMA [45] have shown promise in significantly reducing the burden on programmers by being pre-trained on extensive code datasets. However, their efficacy in tasks concerning compiler backends, as evidenced by experimental findings in Sec. 4.3, remains limited, indicating ample room for enhancement. Moreover, the compiler community currently lacks a publicly available large-scale backend dataset, which could enhance the efficiency of backend development across diverse targets.

In this paper, we present ComBack, which is the first public dataset leading to a promising future for the application of language models for backend development. ComBack comprises 178 backends for mainstream compilers (77 from GCC and 101 from LLVM), sourced from open-source GitHub repositories. We also design three tasks to evaluate the performance of language models based on ComBack for three prevalent scenarios encountered in backend development, including 1) Statement-Level Completion, 2) Next-Statement Suggestion, 3) Code Generation.

In the experiment, we selected 6 representative open-source language models [50, 23, 14, 49, 22, 7] and fine-tuned them with ComBack. The results indicate that ComBack effectively improves the accuracy of 6 language models across 3 tasks. Furthermore, we conducted research on executing three code tasks for three new targets within GCC and LLVM. Additionally, experimental findings show that fine-tuning a model with just 220M parameters based on ComBack significantly boosts programmers' efficiency compared to Fork-Flow, ChatGPT and Code-LLaMA, demonstrating the value of ComBack in enhancing the language model's performance with compiler backend development.

## 2   Background: Conventional Backend Development Process

To develop a compiler backend for a new target, programmers are required to provide specific implementations for a series of compiler infrastructure provided function interfaces based on target-dependent information and characteristics, including instruction sets, registers, byte order, and similar attributes. Specifically, functions within a backend can be divided into two categories:

**Inherited Functions.** This category includes compiler infrastructure function interfaces that carry out specific tasks in the backend process. Programmers must inherit these interfaces and provide implementations tailored to each target. For instance, the "getReloctype" function in LLVM maps relocation variants and immediate values in instruction sets. Differences in this function across targets mainly involve target-specific relocation variants and immediate values. It's important to note that programmers need not to implement all provided interfaces but only a subset relevant to the target, resulting in variations in the implemented inherited functions across different targets.

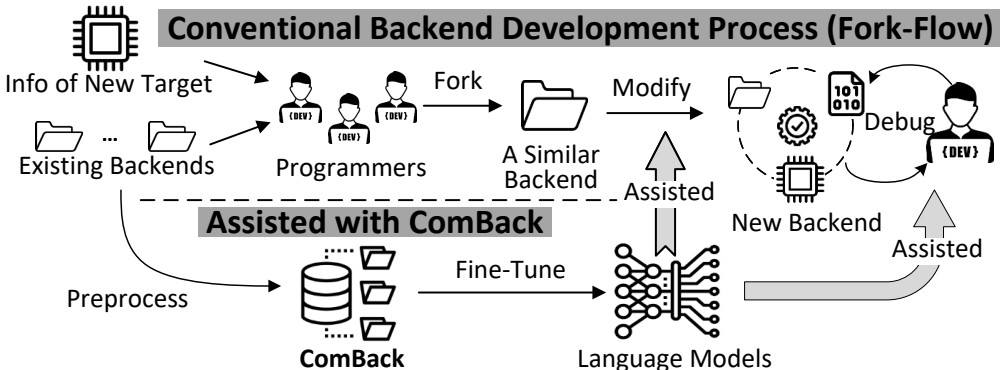

Figure 3: Conventional backend development process and assisted process with ComBack.

**Customized Functions.** This category includes specialized functions designed specifically for certain targets. For example, the "islmm24bit" function in ARM target checks if the encoding length of an immediate value is 24 bits, unique to ARM and not found in other targets.

Fig. 2 shows the proportion of two types of functions in LLVM 17.0.1. In BPF, 82.3% of functions, and in X86, 66.3%, are inherited from LLVM interfaces. On average across all 25 targets, inherited functions account for 57.6%. This prevalence highlights the significant presence of inherited functions across various targets, indicating a notable commonality among them.

Fig. 3 depicts the conventional backend development process (**Fork-Flow**) [43, 32], where programmers must acquire knowledge of the unique characteristics of a new target, such as instruction formats and target-specific flags. They then fork an existing backend that shares similarities (e.g., both being CPU or GPU) and make modifications based on this knowledge to create a tailored backend for the new target. Despite its steep learning curve, similarities among backends of the same type result in redundant development efforts, causing inefficiencies in manual work.

To mitigate this challenge, we propose ComBack, which can be utilized to fine-tune models and facilitate fine-tuned models to assist programmers with backend development, as shown in Fig. 3, thereby reducing redundant efforts and enhancing efficiency.

## 3 ComBack: A Dataset for Compiler Backend Development

### 3.1 Overview of ComBack

To the best of our knowledge, ComBack is the first public dataset for compiler backend development. Notably, it comprises three features as outlined below:

(1) **Large-Scale.** ComBack is sourced from 317 GitHub repositories and the official websites of GCC [20] and LLVM [33], covering versions 3.0 to 13.0 for GCC and 2.0.1 to 17.0.1 for LLVM. It includes 43,299 functions and 883.7 KLoC (Kilo lines of code) for GCC, and 138,940 functions and 4,847.5 KLoC for LLVM, shown in Table 1. Its large scale enhances model performance on common backends and facilitates generalization to less common ones.

(2) **Multi-Target.** Mainstream compiler infrastructure now supports multiple backends for diverse targets, requiring ComBack to be inclusive of such diversity. As indicated in Table 1, there are a total of 77 targets for GCC backends and 101 for LLVM backends in ComBack. These targets cover various types including CPUs, MPUs (Micro-Processors), GPUs, etc. Among them, CPUs and MPUs are more abundant due to their wide applicability across various scenarios. In contrast, other types of processors such as GPUs and DSPs are fewer as they are usually designed for specific tasks, such as GPUs for deep learning workloads and parallel data computation. Leveraging commonalities among these targets, as discussed in Sec. 1, enables models to learn cross-target patterns, facilitating advanced research among various backends. For detailed target information, refer to Appendix A.

Table 1: Data statistics about targets and code in ComBack.

(a) GCC

| Type | Target | Function | KLoC |
|---|---|---|---|
| CPU | 30 | 35,147 | 647.2 |
| MPU | 33 | 6,010 | 183.9 |
| GPU | 2 | 457 | 11.2 |
| VLIW | 5 | 959 | 25.4 |
| DSP | 3 | 399 | 9.6 |
| Virtual | 4 | 327 | 6.5 |
| **Sum** | **77** | **43,299** | **883.7** |

(b) LLVM

| Type | Target | Function | KLoC |
|---|---|---|---|
| CPU | 43 | 84,914 | 3,450.4 |
| MPU | 30 | 11,311 | 173.0 |
| GPU | 5 | 22,591 | 768.3 |
| VLIW | 4 | 2,048 | 24.3 |
| DSP | 7 | 9,646 | 263.2 |
| Virtual | 12 | 8,430 | 168.3 |
| **Sum** | **101** | **138,940** | **4,847.5** |

(3) **Versatility.** To tackle real-world challenges in compiler backend development, like code completion, ComBack focuses on enhancing model versatility. It covers three tasks: 1) Statement-Level Completion; 2) Next-Statement Suggestion; 3) Code Generation, aiding programmers in backend modification and customization. By analyzing diverse target backends, models can better assist with code completion and generation for both existing and new backends. This adaptable approach reduces programming workload, enabling ComBack to handle various scenarios.

## 3.2 Data Collection and Pre-processing

The collection and pre-processing of data in ComBack adhere to the following steps:

1. **Code Collection.** We crawled GitHub using "GCC/LLVM+Backend" as keywords, filtering out incomplete repositories. This yielded 21 GCC repositories and 296 LLVM repositories. We also collected source code versions 3.0 to 13.0 from the official GCC website [20], and versions 2.0.1 to 17.0.1 from the official LLVM website [33]. The backend code from multiple repositories was aggregated and reorganized by targets to create the raw code data.

2. **Function Description Collection.** We collected function descriptions from two sources. Firstly, we extracted descriptions directly from comments within the source code associated with each function. Additionally, for LLVM, we obtained function descriptions from its official Doxygen website [31] using crawling techniques to analyze them further.

3. **Code Extraction.** We started by removing duplicate files and comments from the source code for each target to minimize their influence on fine-tuning. Then, we used the tree-sitter tool [47] to extract functions from the code after comment removal. Each line ending with ";", ":", "{", or "}" was partitioned into a single statement, allowing us to obtain all functions within the backend source code along with their internal statements.

4. **Target-Specific Value Extraction.** Backend code, unlike basic C/C++ programs, prominently includes target-specific values comprising information and characteristics of the instruction set architecture (ISA) of the corresponding target. Fig. 4(a)-(c) illustrates three typical target-specific values: instruction encodings (Fig. 4(c)), size (Fig. 4(c)), immediate values (Fig. 4(b)), and target-specific flags (Fig. 4(a)).

   Observations indicate that target-specific values can be categorized into 3 types: (1) numerical values (Fig. 4(b) and (c)); (2) strings in double quotation marks (Fig. 4(c)); (3) enumeration variable values with the target's name prefix (Fig. 4(a)). However, some enumeration values may start with the target name abbreviation, like "PPC" for `PowerPC`.

   We use a script to automatically filter out target-specific values based on these patterns, including enumeration values starting with abbreviations, like "PPC". Following the approach used in CodeXGlue [34], we replace target-specific values in the code with intermediate representations: "<ISA_LIT>" for enumeration variables, "<NUM_LIT>" for numerical values, and "<STR_LIT>" for strings. Moreover, we store each target-specific value corresponding to these intermediate representations. All target abbreviations are listed in Appendix B.

```
case RISCVII::MO_LO:                      ...                    ...
   Kind = RISCVMCExpr::VK_RISCV_LO;       return isImm(16, 31);  OS.write("\0\0\x40\x03", 4);

case CSKYII::MO_GOT32:                     ...                    ...
   Kind = CSKYMCExpr::VK_CSKY_GOT;        return isImm(−8, 7);   OS.write("\x20", 1);
```

(a) Target-Specific Flag and VariantKind  (b) Immediate Value  (c) Instruction Encoding and Size

Figure 4: Examples of target-specific values in GCC and LLVM.

**Inputs:** ... adjustReg(DL, SPReg, FPReg, −StackSize+RVFI−>getVarArgsSaveSize(), ______________
**Ground Truth:** MachineInstr::FrameDestroy);

(a) Statement-Level Completion

**Inputs:** ... maxCallFrameSize = (maxCallFrameSize + AlignMask) & ~AlignMask;
**Ground Truth:** MFI −> setMaxCallFrameSize(maxCallFrameSize);

(b) Next-Statement Suggestion

**Inputs:**
getPointerRegClass: Returns a TargetRegisterClass used for pointer values.
Target−Specific Value: Sparc, SP::I64RegsRegClass, SP::IntRegsRegClass.
**Ground Truth:**
TargetRegisterClass ∗SparcRegisterInfo::getPointerRegClass(MachineFunction &MF ,unsigned Kind) {
     return Subtarget.is64Bit() ? &SP::I64RegsRegClass : &SP::IntRegsRegClass;
}

(c) Code Generation

Figure 5: Examples of three tasks in ComBack.

## 3.3 Tasks in ComBack

For two common scenarios in compiler backend development, we've outlined three tasks, depicted in Fig. 5. For on-the-fly programming, we've devised Statement-Level Completion (Fig. 5(a)) and Next-Statement Suggestion (Fig. 5(b)) [37], aiming to speed up the programming process. For situations where programmers provide function descriptions in natural language, we've introduced Code Generation (Fig. 5(c)), facilitating direct code generation for a given function. Data processing steps for each task are detailed in subsequent subsections.

Language models fine-tuned with ComBack aid programmers in backend development by completing current statements (Statement-Level Completion), predicting next statements (Next-Statement Suggestion) based on the contextual information. Additionally, it can generate functions based on provided natural language descriptions and target-specific values (Code Generation), reducing repetitive tasks and enhancing efficiency.

### 3.3.1 Statement-Level Completion

Following the data extraction method used in the code completion dataset of CodexGlue [34], we initially aimed to extract five consecutive statement sequences randomly from each function in every backend. We retained samples where the proportion of tokens in the sequence relative to the entire function exceeded 30%, aiming to capture more contextual semantics. Assuming each sample contains $n$ statements, we used the first $n − 1$ statements along with 50%-90% of tokens from the $n_{th}$ statement as input. The remaining 10%-50% of tokens from the $n_{th}$ statement served as ground truth, with this ratio chosen randomly. We treated tokens like ";", ":", "{", "}" in C/C++ as statement terminators, as described in Sec. 3.2. We maintained the intermediate representations from Sec. 3.2 in the task's input and ground truth because target-specific values are sourced from ISA of the target,

making accurate prediction based solely on the code context challenging. Finally, we filtered out data with input lengths exceeding 512 tokens or output lengths exceeding 128 tokens, resulting in a total of 161,124 samples for Statement-Level Completion.

### 3.3.2 Next-Statement Suggestion

Data processing for Next-Statement Suggestion mirrors that of Statement-Level Completion. We randomly extract five consecutive statement sequences from each function in every backend, retaining samples with over 30% of the function's tokens. The main distinction is that, for a Next-Statement Suggestion sample with $n$ statements, the preceding $n - 1$ statements serve as input, while the $n_{th}$ statement serves as the ground truth, as shown in Fig. 5(b). We also retained the intermediate representation in code and filtered out samples with input lengths exceeding 512 tokens or ground truth lengths surpassing 128 tokens. Finally, we obtained the dataset comprising 216,315 samples for Next-Statement Suggestion.

### 3.3.3 Code Generation

For Code Generation, we only kept functions with natural language descriptions (68.08% functions in LLVM and 48.12% functions in GCC), discarding those lacking such descriptions. Each function's description, along with its internal target-specific values, was used as input (typically requiring extraction from ISA manuals), while the entire function (replacing each intermediate representation with corresponding target-specific value) served as the ground truth, as seen in Fig. 5(c). During filtering, samples with input exceeding 256 tokens or ground truth surpassing 512 tokens were removed, retaining 45,296 samples.

## 4 Experiment

This section addresses the following research questions:

- **RQ.1**: Can ComBack effectively enhance backend development capabilities of various language models? (Sec. 4.2)

- **RQ.2**: Can ComBack facilitate fine-tuning a model to enhance backend development efficiency for new targets of existing types and new types? (Sec. 4.3)

- **RQ.3**: Can ComBack support iterative expansion to improve backend development efficiency for customized targets? (Sec. 4.4)

### 4.1 Experimental Setup

**Fundamental Models.** We selected six open-source language models pre-trained or fine-tuned on C or C++ language: 1) CodeBert (Fine-Tuned with C) [14, 16], 2) GraphCodeBert (Fine-Tuned with C) [23, 15], 3) UniXcoder-base-nine [22], 4) CodeT5-base [50], 5) CodeT5+-220M [49] and 6) NatGen [7]. We chose them for two reasons: 1) these models are representative open-source programming language models, suitable for various tasks in ComBack; 2) their relatively small model size helps reduce computational resources needed for training and deployment. All fine-tuned models and code are available at https://huggingface.co/docz1105/ComBack_Models.

**Baselines.** For experiment in Sec. 4.3, we include Fork-Flow method as the baseline of conventional development efficiency. Additionally, we choose ChatGPT-3.5-Turbo and Code-LLaMA-34B-Instruct as baselines for mainstream large language models (LLMs). ChatGPT is the most widely used LLM globally, while Code-LLaMA, an open-source LLM designed specifically for code-related tasks, achieves state-of-the-art performance on many code related benchmarks.

**Evaluation Metrics.** To evaluate the inference capability of models fine-tuned with ComBack, we use exact match accuracy (EM) and Levenshtein Edit Distance Similarity (ED) [22, 34] for Statement-Level Completion and Next-Statement Suggestion. For Code Generation, we use Levenshtein Edit Distance Similarity and BLEU-4 [38] as evaluation metrics.Exact Match was used for the two code completion tasks because it directly measures the correctness of the generated code, meeting developers' needs in real-time programming. For Code Generation, we chose BLEU-4 to assess structural similarity between the generated code and the ground truth, the higher the BLEU-4 score,

Table 2: Comparison of accuracy across three tasks of six models fine-tuned by ComBack.

| Model | Stmt. Comp. | | Next. Sugg. | | Code. Gen. | | Stmt. Comp. | | Next. Sugg. | | Code. Gen. | |
|---|---|---|---|---|---|---|---|---|---|---|---|---|
| | EM (%) | ED | EM (%) | ED | BLEU-4 | ED | EM (%) | ED | EM (%) | ED | BLEU-4 | ED |
| | **Without Fine-Tuning** | | | | | | **Fine-Tuned** | | | | | |
| CodeBert | 0.00 | 0.97 | 0.00 | 1.31 | 0.00 | 0.44 | 53.84 | 77.44 | 52.67 | 70.82 | 23.54 | 54.63 |
| GraphCodeBert | 0.00 | 0.35 | 0.00 | 0.54 | 0.00 | 2.41 | 43.00 | 71.89 | 47.10 | 61.31 | 20.73 | 48.83 |
| UniXcoder | 0.07 | 27.56 | 15.93 | 29.11 | 0.00 | 31.81 | **67.84** | **85.06** | 58.51 | 75.31 | 56.24 | 73.45 |
| CodeT5 | 0.65 | 21.45 | 7.23 | 23.50 | 0.00 | 13.57 | 66.47 | 84.34 | 58.52 | 76.03 | 70.87 | 80.45 |
| NatGen | 0.00 | 13.52 | 0.02 | 15.95 | 0.01 | 28.76 | 67.47 | 84.83 | **60.30** | **76.84** | 71.73 | 81.39 |
| CodeT5+ | 0.02 | 7.24 | 0.12 | 9.87 | 0.00 | 12.33 | 66.93 | 84.45 | 59.57 | 76.41 | **75.29** | **82.92** |

the greater the similarity. We also used edit distance for all tasks to measure the modifications needed to align the generated code with the ground truth, where a higher score indicates fewer required edits and closer alignment to the ground truth.

**Training Settings.** All models are trained and evaluated on a server with a 64-core Intel Xeon Gold CPU and 8 NVIDIA Tesla V100 GPUs, each with 16GB of memory. We set the fine-tuning objective as: sequence-to-sequence prediction for three tasks. To ensure fairness, all hyperparameters are identical for the six models, detailed in Appendix C.

## 4.2 Accuracy Improvement across Various Models

To evaluate accuracy improvement of different models across three tasks, we randomly split the backend data from all targets into train/validation/test sets in an 80%:10%:10% ratio, with details on the quantity of data and tokens in each set provided in Appendix D. Subsequently, we fine-tuned and tested six models with the dataset. Table 2 shows the accuracy improvement of six models across three tasks after fine-tuning with ComBack. The models exhibited improvements of 41.64 - 77.21 of ED across three tasks, 42.58%-67.77% in absolute terms of EM for Statement-Level Completion and Next-Statement Suggestion, and 20.73-75.29 of BLEU-4 for Code Generation.

**Answer to RQ.1: ComBack can effectively improve backend development capabilities of various language models.**

## 4.3 Efficiency Enhancement for New Targets

In Sec. 4.3.1 and Sec. 4.3.2, we simulate code completion and generation scenarios for new targets of existing types and new types. We select CodeT5+ for experiments in following sections, as it achieves the highest accuracy on average across three tasks (Sec. 4.2).

### 4.3.1 Targets of Existing Types

We simulate code completion and generation scenarios for new targets of existing types. Therefore, we select `RISC-V` (CPU), `ARC` (MPU), and `NVPTX` (GPU) in GCC and LLVM as test sets. Other CPU, MPU, and GPU targets are split into train and validation sets at an 85%:15% ratio. `RI5CY` in LLVM is excluded since it's a customized target based on `RISC-V` and shares most code with it. Further dataset details are provided in Appendix D.

Next, we fine-tuned CodeT5+ with the dataset including CPU, MPU and GPU. We further compared the accuracy of fine-tuned CodeT5+, mainstream LLMs, and conventional backend development methods (Fork-Flow) for backend development of three targets.

**Mainstream LLMs.** We evaluated the performance of ChatGPT-3.5-Turbo and Code-LLaMA-34B-Instruct across three tasks for `RISC-V`, `ARC`, and `NVPTX`, as shown in Table 3. Inputs for both LLMs closely matched those in ComBack, with the addition of a unified prompt, detailed in Appendix F.

CodeT5+ consistently outperforms two LLMs across three tasks. Specifically, in Statement-Level Completion, CodeT5+ surpasses 37.10%-40.82% for EM compared with ChatGPT and 49.72%-54.57% compared with Code-LLaMA in absolute terms on three targets in GCC and LLVM. The significant improvement in accuracy indicates that *fine-tuning small LLMs with ComBack exceeded large LLMs significantly*. Therefore, ComBack holds significant importance in enhancing the performance of language models in backend development scenarios.

Table 3: Accuracy of code generated by ChatGPT, Code-LLaMA and CodeT5+ fine-tuned by ComBack for targets of existing types.

| Model | Stmt. Comp. | | | | | | Next. Sugg. | | | | | | Code. Gen. | | | | | |
|---|---|---|---|---|---|---|---|---|---|---|---|---|---|---|---|---|---|---|
| | RISC-V | | ARC | | NVPTX | | RISC-V | | ARC | | NVPTX | | RISC-V | | ARC | | NVPTX | |
| | EM (%) | ED | EM (%) | ED | EM (%) | ED | EM (%) | ED | EM (%) | ED | EM (%) | ED | BLEU-4 | ED | BLEU-4 | ED | BLEU-4 | ED |
| GCC | | | | | | | | | | | | | | | | | | |
| ChatGPT | 10.34 | 38.41 | 15.35 | 42.94 | 12.01 | 41.47 | 6.44 | 12.90 | 9.75 | 20.79 | 7.97 | 17.79 | 1.37 | 24.12 | 1.67 | 28.26 | 1.57 | 26.97 |
| Code-LLaMA | 0.41 | 19.07 | 0.85 | 16.77 | 0.56 | 18.22 | 1.58 | 13.54 | 2.66 | 17.95 | 2.47 | 16.59 | 1.67 | 27.89 | 1.71 | 30.49 | 1.57 | 27.65 |
| **CodeT5+** | **51.16** | **75.32** | **52.45** | **74.57** | **50.56** | **75.52** | **49.11** | **67.84** | **38.26** | **59.21** | **38.33** | **56.31** | **32.56** | **58.67** | **19.94** | **50.27** | **25.47** | **52.60** |
| LLVM | | | | | | | | | | | | | | | | | | |
| ChatGPT | 12.08 | 41.39 | 16.77 | 42.02 | 14.73 | 43.72 | 9.80 | 21.86 | 10.81 | 20.66 | 11.39 | 22.82 | 1.23 | 25.12 | 1.30 | 27.19 | 1.43 | 25.45 |
| Code-LLaMA | 0.45 | 17.61 | 0.61 | 17.21 | 0.99 | 17.23 | 1.75 | 15.04 | 0.42 | 11.27 | 2.42 | 16.25 | 1.43 | 27.24 | 1.61 | 32.12 | 1.59 | 28.08 |
| **CodeT5+** | **62.68** | **82.02** | **71.34** | **85.98** | **64.45** | **81.53** | **48.71** | **68.95** | **58.68** | **74.57** | **47.81** | **65.51** | **50.34** | **72.98** | **55.38** | **74.41** | **44.33** | **66.36** |

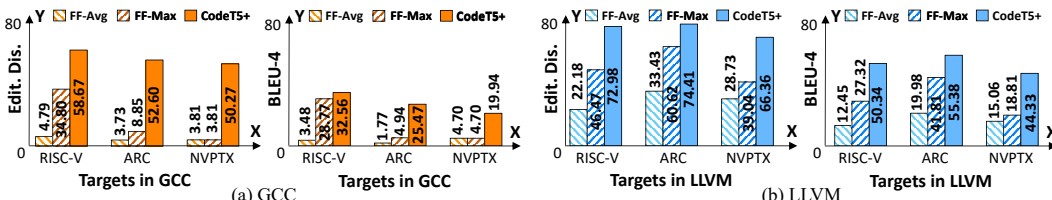

Figure 6: Comparison of fine-tuned CodeT5+ and Fork-Flow for Code Generation, where "**FF**" is the abbreviation of Fork-Flow.

**Fork-Flow.** Due to the similarity between the Fork-Flow process, which involves modifying complete functions, and scenarios in Code Generation where developers modify functions automatically generated by the model, we only compare Fork-Flow with fine-tuned CodeT5+ on Code Generation.

To simulate the process of Fork-Flow, we used scripts to calculate the ED and BLEU-4 between functions with identical names on new targets (RISC-V, ARC, NVPTX) and their corresponding implementations on other targets. We aggregate their average and maximum values across these targets (excluding RISC-V, ARC, NVPTX) and compare them with values of functions generated by fine-tuned CodeT5+, as depicted in Fig. 6. It is evident that the accuracy of fine-tuned CodeT5+ exceeds both the average and maximum values of Fork-Flow, demonstrating that the CodeT5+ fine-tuned by ComBack can achieve higher efficiency compared to conventional development method. Details of Fork-Flow can be viewed in Appendix E.

### 4.3.2 Targets of New Types

We further explore whether ComBack can facilitate code completion and generation for targets of new types. We fine-tune CodeT5+ with CPU data only, excluding MPU and GPU data from train and validation sets in Sec. 4.3.1. Next, we exclude CPU data and only retain MPU and GPU data in the test set, detailed in Appendix D. After fine-tuning CodeT5+ with the dataset only containing CPU, we explore whether it can generate functions for new types of targets (MPU and GPU) in the test dataset.

Results in Table 4 indicate that CodeT5+ fine-tuned on existing types of targets (CPU) can indeed facilitate code completion and generation for new types of targets (MPU and GPU), as backends of different types of targets under the same compiler infrastructure (GCC or LLVM) adhering to unified programming standards (such as same function interfaces and classes).

However, there tends to be a decrease in accuracy on most targets, as depicted in Table 4. Further analysis in Appendix H reveals that there are differences in functions required in the backend of different types of targets. Therefore, the fine-tuned model struggles to effectively complete and generate code corresponding to some functions for new types of targets.

**Answer to RQ.2: The model fine-tuned by ComBack can enhance backend development efficiency for new targets of both existing and new types.**

### 4.4 Iterative Expansion Ability

In this section, we explore ComBack's iterative expansion ability. As application scenarios diversify, the field of processor design witnesses a proliferation of customized targets. These targets, often

Table 4: Accuracy across three tasks of targets of new types (MPU and GPU).

| Dataset | Stmt. Comp. | | | | Next. Sugg. | | | | Code. Gen. | | | |
|---|---|---|---|---|---|---|---|---|---|---|---|---|
| | ARC (MPU) | | NVPTX (GPU) | | ARC (MPU) | | NVPTX (GPU) | | ARC (MPU) | | NVPTX (GPU) | |
| | EM (%) | ED | EM (%) | ED | EM (%) | ED | EM (%) | ED | BLEU-4 | ED | BLEU-4 | ED |
| GCC | | | | | | | | | | | | |
| -w/o GPU and MPU | 50.53 | 74.09 | 46.37 | 72.45 | 37.22 | 58.21 | 38.33 | 56.83 | 19.29 | 49.12 | 22.46 | 50.33 |
| -w GPU and MPU | 52.45 | 74.57 | 50.56 | 75.52 | 38.26 | 59.21 | 38.33 | 56.31 | 19.94 | 50.27 | 25.47 | 52.60 |
| **Diff** | **-1.92** | **-0.48** | **-4.19** | **-3.07** | **-1.04** | **-1.00** | **0.00** | **+0.52** | **-0.65** | **-1.15** | **-3.01** | **-3.37** |
| LLVM | | | | | | | | | | | | |
| -w/o GPU and MPU | 69.82 | 85.59 | 60.04 | 79.85 | 58.26 | 73.75 | 46.28 | 63.92 | 49.62 | 70.26 | 42.94 | 65.43 |
| -w GPU and MPU | 71.34 | 85.98 | 64.45 | 81.53 | 58.68 | 74.57 | 47.81 | 65.5 | 55.38 | 74.41 | 44.33 | 66.36 |
| **Diff** | **-1.52** | **-0.39** | **-4.41** | **-1.68** | **-0.42** | **-0.82** | **-1.53** | **-1.58** | **-5.76** | **-4.15** | **-1.39** | **-0.93** |

Table 5: Improvement of accuracy across three tasks for `RI5CY` after iterative expansion.

| Dataset | Stmt-Level. Comp. | | Next-Stmt. Sugg. | | Code. Gen. | |
|---|---|---|---|---|---|---|
| | EM (%) | ED | EM (%) | ED | BLEU-4 | ED |
| -w RISC-V | 74.06 | 87.91 | 67.25 | 81.28 | 79.46 | 89.92 |
| -w/o RISC-V | 66.16 | 83.79 | 57.29 | 74.73 | 54.41 | 75.41 |
| **Diff** | **+7.90** | **+4.12** | **+9.96** | **+6.55** | **+25.05** | **+14.51** |

built upon existing targets, integrate customized instructions to swiftly cater to specific application scenarios. Consequently, their backends merely require extensions from the existing backend. We chose `RI5CY` in LLVM to test if ComBack can be iteratively expanded to improve backend development efficiency for customized targets.

As a target based on `RISC-V`, `RI5CY` shares most backend code with `RISC-V` but includes customized instruction handling. Initially, we fine-tuned CodeT5+ with train and validation set in Sec. 4.3.2 (excluding `RISC-V`), then we add `RISC-V` into train and validation set and fine-tuned CodeT5+ with new data (detailed in Appendix D) and restart fine-tuning from scratch. Results in Table 5 show a notable accuracy improvement across three tasks after integrating `RISC-V` data, demonstrating ComBack's iterative expansion ability.

**Answer to RQ.3: ComBack effectively enables backend development for customized targets by iterative data expansion.**

# 5 Related Work

**Backend Development.** Compiler backend development heavily relies on manual efforts. Some researchers have proposed Processor Design Languages (PDL) to describe ISA and hardware information for processors [40, 6, 13, 4, 12, 24, 35, 5]. While these methods mitigate manual efforts to some degree, programmers still need to invest significant effort in learning PDL rules and writing files.

**Dataset for Compiler.** Datasets like CodeXGlue [34] and CodeSearchNet [25] have enhanced language models in programming. As AI extends into compilers, datasets like Compile [21], TenSet [56], and ANGHABENCH [11] focus on compiler optimization. However, there remains a dearth of datasets tailored for compiler backends within the community. ComBack is the first dataset designed to substantially augment the capabilities of language models in backend code generation.

**AI for Compilation.** AI has driven the widespread adoption of machine-learning-based compilation techniques. These methods have found application in tasks such as developing cost and performance models [54, 44, 55, 42, 36], determining transformation order [48, 17, 39, 29, 8], and optimizing parallel programs [28, 27, 52, 51, 26]. Ongoing projects using transformer models for decompilation[1, 2, 46, 53] and code optimization [10] highlight the significant potential of AI for compilers.

# 6 Discussion

**Limitation.** One limitation of ComBack is the absence of function descriptions for highly-customized functions in backends for specific targets. We plan to address this in future iterations of the dataset.

**Potential Societal Impact.** ComBack does not contain any personally identifiable information or offensive content, thereby mitigating any potential negative societal impact.

**Conclusion.** In this paper, we introduce ComBack, the first public dataset for compiler backend development. ComBack includes 178 backends for mainstream compilers and features three tasks, including statement-level completion, next-statement suggestion and code generation. It enables efficient backend code completion and generation after fine-tuning language models with ComBack. Our evaluation, conducted on six representative language models, shows that ComBack boosts language models' performance across all three tasks. Notably, CodeT5+ with only 220M parameters significantly outperforms the efficiency of conventional backend development methods and even surpasses ChatGPT-3.5-Turbo and Code-LLaMA-34B-Instruct across three tasks, suggesting potential improvements in compiler development speed and efficiency.

## Acknowledgement

We would like to thank all anonymous reviewers for their insightful feedback. This work was supported by National Key R&D Program of China, Grant No. 2023YFB3001502, the Strategic Priority Research Program of the Chinese Academy of Sciences, Grant No.XDB0500102 and XDB0660102. It was also supported by the National Natural Science Foundation of China, Grant No.U23B2020, No. 62090024, No. 62302479 and the Innovation Funding of ICT, CAS under Grant No.E361010 and No.E261110.

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

# A   Appendix: Target List in ComBack.

In Table 6, we provide all targets in ComBack.

Table 6: Target list in ComBack.

| Compiler | ISA | Target |
|---|---|---|
| GCC | CPU | aarch64, arm, clipper, crx, csky, d30v, i370, i386, i860, i960, ia64, iq2000, loongarch, mep mips, mmix, moxie, mt, nds32, or1k, pa, powerpcspe, pru, riscv, rs6000, rx, sh, sparc stormy16, vax, bfin, c4x, fr30, gcn, nvptx |
| | MPU | 1750a, a29k, alpha, arc, avr, cr16, cris, eco32, epiphany, ft32, h8300, lm32, m32c, m32r m68hc11, m68k, m88k, mcore, microblaze, mn10200, mn10300, msp430, nios2, ns32k pdp10, pdp11, rl78, romp, s390, spu, v850, xtensa, z8k |
| | Virtual | bpf, mapip, visium, vms |
| | VLIW | c6x, convex, frv, tilegx, tilepro |
| LLVM | CPU | AArch64, ARM, ARM64, AZPR, CAHP, CJG, Comet2, Cpu0, CSKY, Dcpu16, Digital DLX, F2003f, FISC, FPGA, IA64, Kudeyar, Lanai, LC2200, LC3, LC3b, LEG LoongArch, Mandarin, MINA32, Mips, MMIX, OpenRISC, OR1K, PowerPC, RI5CY RISCV, SHUXI, SIC, Sparc, StackPU2, SystemZ, TeeRISC, TOY, UPT, VE, X86, XNCM |
| | DSP | Blackfin, Hexagon, MDSP, SNES, Teak, Videocore, VideoCore4 |
| | GPU | AMDGPU, NVPTX, Nyuzi, PTX, R600 |
| | MPU | AAP, AGC, Alpha, ARC, ARCompact, AVR, CellSPU, ECLair, Epiphany, GBZ80, J2 LM32, M680x0, M68k, M88k, MBlaze, MCS51, MOS, MSP430, Nios2, P2, PIC16 TL45, TLCS900, TriCore, WDC65816, XCore, Xtensa, Z80, Z80old |
| | Virtual | BPF, DirectX, HSAIL, JVM, mproc, NPEngine, RV16K, SPIRV, TGSI, TPC, TVM WebAssembly |
| | VLIW | Patmos, rvex, Tile64, TMS320C64X |

# B   Appendix: Target abbreviation occurred during pre-processing.

Table 7: Targets Abbreviation in ComBack.

| Target | Abbreviation | Target | Abbreviation | Target | Abbreviation |
|---|---|---|---|---|---|
| AMDGPU | SI | ARCompact | ARC | Mandarin | MD |
| Blackfin | BF | CellSPU | SPU | PowerPC | PPC |
| DirectX | DXIL | GBZ80 | GB | R600 | SI |
| RI5CY | RISCV | Sparc | SP | Tile64 | T64 |
| Videocore | VC | WDC65816 | WDC | | |

In Table 7, we provide all abbreviations for targets in ComBack. Recording these abbreviations can assist us in accurately extracting target-specific values.

# C   Appendix: Hyperparameters and Input/Output Sequence Length Settings.

In Table 8, we provide all hyperparameter settings. For CodeBert and GraphCodeBert, the input sequence length is set to 384, with output lengths of 128 for Statement-Level Completion and Next-Statement Suggestion, and 256 for both input and output for Code Generation, given the maximum token length of 512 for both models. For the other four models, the input sequence length is set to 512, with output lengths of 128 for Statement-Level Completion and Next-Statement Suggestion, and 256 for input and 512 for output for Code Generation.

Table 8: Hyperparameter settings.

| Hyperparameter | Value | Hyperparameter | Value | Hyperparameter | Value |
|---|---|---|---|---|---|
| Training Batch Size | 32 | Beam Size | 4 | Learning Rate | 5e-5 |
| Evaluation Batch Size | 16 | Max Optimization Steps | 3 | | |

# D Appendix: Data Statistics about the Number and Token of Three Tasks.

In Table 9, we provide all detailed data of train, validation and test set of experiments in Sec. 4.2 to Sec. 4.4.

Table 9: Data statistics about the number and token of three tasks.

(a) Data statistics about the number and token of three tasks for Sec. 4.2.

| Task | Train | validation | Test |
|---|---|---|---|
| Stmt. Comp. | 128,899(11.36M Token) | 16,112(1.43M Token) | 16,113(1.43M Token) |
| Next. Sugg. | 173,052(15.69M Token) | 21,631(1.99M Token) | 21,632(1.98M Token) |
| Code. Gen. | 36,236(5.10M Token) | 4,530(0.64M Token) | 4,530(0.64M Token) |

(b) Data statistics about the number and token of three tasks for Sec. 4.3.1.

| Task | Train | validation | Test |
|---|---|---|---|
| Stmt. Comp. | 114,016(10.20M Token) | 20,121(1.81M Token) | 6,645(0.58M Token) |
| Next. Sugg. | 152,114(14.10M Token) | 26,844(2.49M Token) | 9,313(0.83M Token) |
| Code. Gen. | 30,633(4.44M Token) | 5,406(0.79M Token) | 2,819(0.37M Token) |

(c) Data statistics about the number and token of three tasks for Sec. 4.3.2.

| Task | Train | validation | Test |
|---|---|---|---|
| Stmt. Comp. | 87,018(7.78M Token) | 15,357(1.37M Token) | 2,764(0.26M Token) |
| Next. Sugg. | 113,684(10.65M Token) | 20,063(1.87M Token) | 4,029(0.38M Token) |
| Code. Gen. | 21,184(3.14M Token) | 3,739(0.55M Token) | 1,372(0.18M Token) |

(d) Data statistics about the number and token of three tasks for Sec. 4.4 (Excluding `RISC-V` in train and validation set).

| Task | Train | validation | Test |
|---|---|---|---|
| Stmt. Comp. | 87,018(7.78M Token) | 15,357(1.37M Token) | 721(0.04M Token) |
| Next. Sugg. | 113,684(10.65M Token) | 20,063(1.87M Token) | 1,035(0.06M Token) |
| Code. Gen. | 21,184(3.14M Token) | 3,739(0.55M Token) | 219(0.02M Token) |

(e) Data statistics about the number and token of three tasks for Sec. 4.4 (Including `RISC-V` in train and validation set).

| Task | Train | validation | Test |
|---|---|---|---|
| Stmt. Comp. | 90,316(8.06M Token) | 15,940(1.42M Token) | 721(0.04M Token) |
| Next. Sugg. | 118,175(11.04M Token) | 20,856(1.94M Token) | 1,035(0.06M Token) |
| Code. Gen. | 22,413(3.30M Token) | 3,957(0.58M Token) | 219(0.02M Token) |

# E    Appendix : Fork-Flow Detailed Experimental Data.

In Table 10, we provide all detailed data in Fork-Flow experiment.

Table 10: Fork-Flow experimental data.

| Compiler | Type | Target | BLEU4 | ED | EM | Target | BLEU4 | ED | EM |
|---|---|---|---|---|---|---|---|---|---|
| GCC | MPU | z8k | 0.32 | 1.33 | 0 | m68k | 1.27 | 2.84 | 0 |
| GCC | MPU | a29k | 0 | 0 | 0 | m88k | 0 | 0 | 0 |
| GCC | MPU | avr | 4.27 | 8.85 | 0.24 | microblaze | 1.39 | 3.53 | 0 |
| GCC | MPU | lm32 | 1.89 | 3.68 | 0.24 | mn10200 | 0 | 0 | 0 |
| GCC | MPU | mcore | 1.4 | 3.61 | 0 | mn10300 | 2.73 | 5.47 | 0 |
| GCC | MPU | msp430 | 0.94 | 1.89 | 0 | nios2 | 3.35 | 7.07 | 0.48 |
| GCC | MPU | v850 | 2.32 | 4.58 | 0 | ns32k | 0 | 0 | 0 |
| GCC | MPU | xtensa | 2.93 | 6.01 | 0.24 | cris | 2.43 | 6.27 | 0 |
| GCC | MPU | cr16 | 1.49 | 3.86 | 0 | pdp11 | 1.39 | 3.75 | 0 |
| GCC | MPU | rl78 | 0.9 | 1.69 | 0 | pdp10 | 0.02 | 0.25 | 0 |
| GCC | MPU | m32c | 1.35 | 4.07 | 0.24 | 1750a | 0 | 0 | 0 |
| GCC | MPU | ft32 | 2.23 | 4.14 | 0 | s390 | 3.53 | 8.05 | 0 |
| GCC | MPU | h8300 | 2.48 | 5.25 | 0 | romp | 0 | 0 | 0 |
| GCC | MPU | alpha | 3.69 | 7.5 | 0.24 | spu | 1.98 | 3.78 | 0 |
| GCC | MPU | epiphany | 4.94 | 7.84 | 0.24 | eco32 | 1.36 | 2.74 | 0 |
| GCC | MPU | m32r | 4.31 | 7.85 | 0.95 | | | | |
| GCC | CPU | aarch64 | 12.54 | 18.21 | 3.51 | sparc | 3.68 | 7.81 | 0.39 |
| GCC | CPU | arm | 4.28 | 7.97 | 0.39 | mep | 0.96 | 2.27 | 0.19 |
| GCC | CPU | csky | 3.77 | 7.76 | 0.19 | vax | 0.78 | 2.13 | 0 |
| GCC | CPU | d30v | 0.19 | 0.49 | 0 | clipper | 0 | 0 | 0 |
| GCC | CPU | i370 | 0 | 0 | 0 | iq2000 | 1.91 | 4.03 | 0.39 |
| GCC | CPU | i386 | 0.26 | 0.68 | 0 | crx | 0.43 | 1.82 | 0 |
| GCC | CPU | i860 | 0 | 0 | 0 | moxie | 1.05 | 2.77 | 0.19 |
| GCC | CPU | i960 | 0 | 0 | 0 | mt | 1.01 | 2.81 | 0 |
| GCC | CPU | ia64 | 2.16 | 5.71 | 0 | nds32 | 1.88 | 4.24 | 0.19 |
| GCC | CPU | loongarch | 28.77 | 34.8 | 8.38 | pru | 2.15 | 5.28 | 0.19 |
| GCC | CPU | mips | 22.24 | 29.99 | 3.51 | rs6000 | 3.41 | 7.25 | 0.19 |
| GCC | CPU | mmix | 1.75 | 4.27 | 0.19 | rx | 1.01 | 2.4 | 0 |
| GCC | CPU | or1k | 2.06 | 4.69 | 0.19 | sh | 2.49 | 5.71 | 0 |
| GCC | CPU | pa | 2.09 | 4.47 | 0 | stormy16 | 0 | 0 | 0 |
| GCC | CPU | powerpcspe | 0.07 | 0.36 | 0 | | | | |
| LLVM | GPU | AMDGPU | 18.81 | 39.04 | 0.58 | PTX | 12.39 | 21.79 | 0.97 |
| LLVM | GPU | Nyuzi | 12.74 | 21.35 | 1.94 | R600 | 16.31 | 32.72 | 0.39 |
| LLVM | MPU | AVR | 28.42 | 45.24 | 2.33 | CellSPU | 11.29 | 25.76 | 0 |
| LLVM | MPU | LM32 | 12.55 | 18.37 | 3.1 | ECLair | 3.94 | 5.4 | 1.55 |
| LLVM | MPU | MCS51 | 28.1 | 43.36 | 2.33 | Epiphany | 0.78 | 0.78 | 0.78 |
| LLVM | MPU | MSP430 | 29.04 | 46.19 | 2.33 | GBZ80 | 27.87 | 45.74 | 0.78 |
| LLVM | MPU | P2 | 28.72 | 42.04 | 4.65 | M680x0 | 24.2 | 39.33 | 4.65 |
| LLVM | MPU | PIC16 | 12.21 | 26.22 | 0 | M68k | 25.49 | 42.28 | 5.43 |
| LLVM | MPU | TriCore | 18.83 | 25.93 | 6.2 | M88k | 23.26 | 41.2 | 5.43 |
| LLVM | MPU | XCore | 41.8 | 60.62 | 5.43 | MBlaze | 15.84 | 29.81 | 0 |
| LLVM | MPU | Xtensa | 22.1 | 41.71 | 6.98 | Nios2 | 12.89 | 20.59 | 2.33 |
| LLVM | MPU | AGC | 13.11 | 22.84 | 3.88 | Z80 | 24.64 | 43.71 | 2.33 |
| LLVM | MPU | TL45 | 24.63 | 38.95 | 5.43 | Z80old | 21.75 | 38.27 | 3.1 |
| LLVM | MPU | TLCS900 | 20.59 | 32.4 | 0 | MOS | 22.77 | 42.36 | 3.1 |
| LLVM | MPU | J2 | 17.75 | 35.71 | 2.33 | AAP | 30.41 | 44.91 | 4.65 |
| LLVM | MPU | Alpha | 12.81 | 25.61 | 0 | WDC65816 | 13.2 | 22.6 | 0 |
| LLVM | MPU | ARCompact | 10.4 | 21.48 | 0 | | | | |
| LLVM | CPU | AArch64 | 27.32 | 46.47 | 1.5 | OR1K | 15.18 | 26.21 | 0.43 |
| LLVM | CPU | ARM | 23.93 | 42.38 | 2.14 | PowerPC | 21.42 | 39.99 | 0.75 |
| LLVM | CPU | ARM64 | 15.33 | 27.04 | 0.75 | SHUXI | 11.21 | 19.73 | 1.71 |
| LLVM | CPU | AZPR | 2.92 | 5.72 | 0.21 | Sparc | 18.19 | 32.98 | 1.61 |
| LLVM | CPU | CAHP | 23.54 | 33.61 | 5.03 | StackPU2 | 2.08 | 2.6 | 0.11 |
| LLVM | CPU | CJG | 11.08 | 19.17 | 1.61 | SystemZ | 21.85 | 38.97 | 1.39 |
| LLVM | CPU | Cpu0 | 16.92 | 29.97 | 1.28 | TOY | 9.45 | 20.55 | 0.32 |
| LLVM | CPU | CSKY | 25.86 | 38.25 | 3.53 | UPT | 5.65 | 12.1 | 0.64 |
| LLVM | CPU | DLX | 12.13 | 24.55 | 1.39 | X86 | 18.88 | 35.77 | 1.39 |
| LLVM | CPU | IA64 | 4.16 | 9.52 | 0 | XNCM | 7.04 | 14.61 | 0.21 |
| LLVM | CPU | Kudeyar | 8.89 | 16.03 | 0.32 | Comet2 | 3.87 | 7.21 | 0.96 |
| LLVM | CPU | Lanai | 16.7 | 30.37 | 1.28 | Dcpu16 | 9.56 | 18.43 | 0 |
| LLVM | CPU | LC2200 | 15.08 | 24.3 | 1.71 | F2003f | 9.24 | 16.72 | 0.54 |
| LLVM | CPU | LC3 | 8.49 | 17.79 | 0.86 | SIC | 11.87 | 22.42 | 1.18 |
| LLVM | CPU | LC3b | 3.14 | 6.47 | 0.32 | TeeRISC | 8.39 | 15.64 | 0.32 |
| LLVM | CPU | LoongArch | 13.6 | 21.83 | 2.57 | Digital | 0.87 | 1.1 | 0.21 |
| LLVM | CPU | Mandarin | 9.24 | 16.75 | 0.54 | FISC | 12.9 | 25.27 | 0.96 |
| LLVM | CPU | MINA32 | 9.96 | 19.04 | 1.07 | FPGA | 1.07 | 2.01 | 0.21 |
| LLVM | CPU | Mips | 22.96 | 40.35 | 2.25 | LEG | 8.94 | 18.63 | 0.86 |
| LLVM | CPU | MMIX | 16.42 | 25.04 | 3.75 | VE | 20.54 | 34.91 | 2.57 |
| LLVM | CPU | OpenRISC | 4.58 | 9.06 | 0.21 | | | | |

# F Appendix: Prompt Example of Input for ChatGPT and Code-LLaMA.

We provide prompt examples of Input for ChatGPT and Code-LLaMA in Fig. 7.

> *//Prompt: Complete the last statement of this code snippet:*
> *...*
> adjustReg(MBB,LastFrameDestroy, DL, SPReg, FPReg, −StackSize+RVFI−>getVarArgsSaveSize()

(a) Statement-Level Completion

> *//Prompt: Predict the next statement of this code snippet:*
> *...*
> maxCallFrameSize = (maxCallFrameSize + AlignMask) & ~AlignMask;

(b) Next-Statement Suggestion

> *//Prompt: Create a function named "getPointerRegClass" for "Sparc" backend of LLVM Compiler.*
> *//The description of this function is "Returns a TargetRegisterClass used for pointer values".*
> *//It contains "Sparc", "SP::I64RegsRegClass", "SP::IntRegsRegClass" as target specific values.*

(c) Code Generation

Figure 7: Prompt examples of tasks in ComBack.

# G Appendix : License of Assets.

In Table 11, we provide all license of assets in experiment.

Table 11: License of assets.

| Assets | CodeBase | License |
|---|---|---|
| CodeBER [14] | CodeSearchNet [25] | MIT License |
| GraphCodeBERT [23] | CodeSearchNet [25] | MIT License |
| UnixCoder [22] | CodeSearchNet [25], C4 [41] | MIT License |
| CodeT5 [50] | CodeSearchNet [25], BigQuery1 [3] | Apache-2.0 |
| NatGen [7] | CodeSearchNet [25], BigQuery1 [3] | MIT License |
| CodeT5+ [49] | GitHub-Code Dataset [9] | bsd-3-clause |

# H Appendix: Heatmap Analysis.

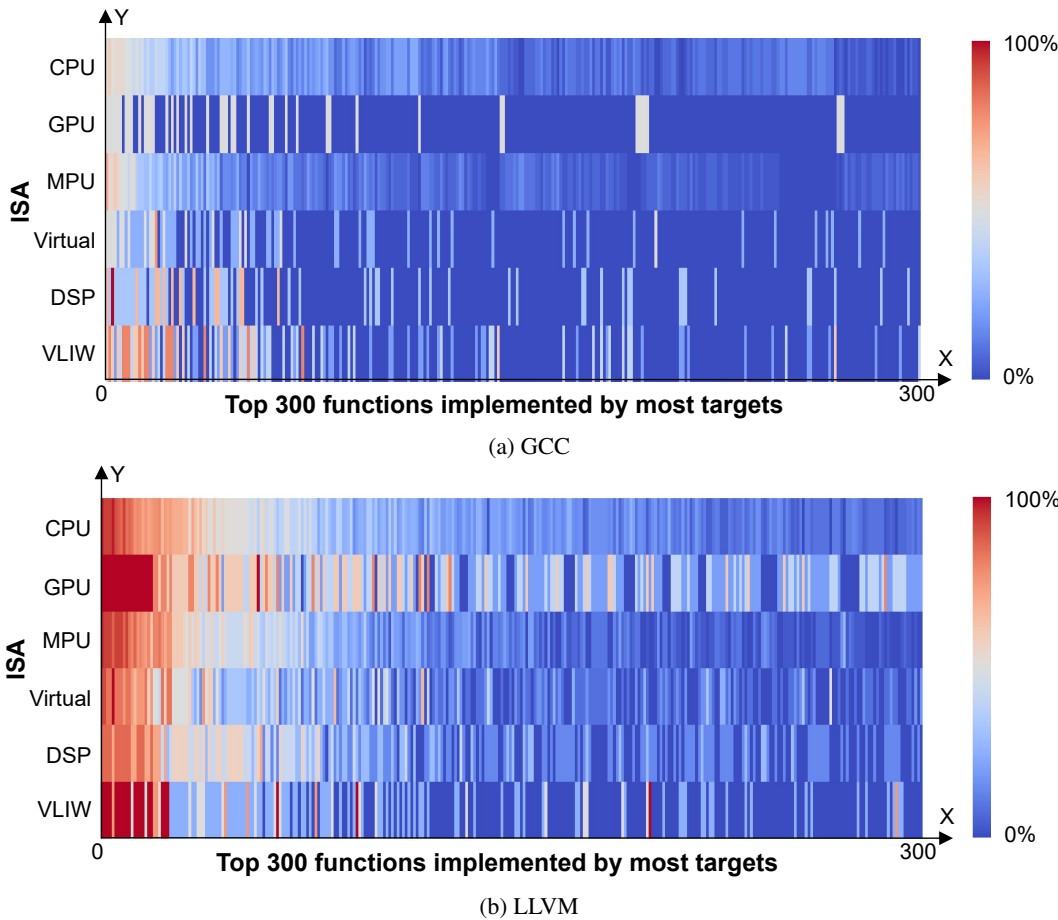

(a) GCC

(b) LLVM

Figure 8: Heatmap analysis of top 300 functions implemented by most targets

We analyzed the top 300 functions implemented by most targets in both GCC and LLVM backends, creating Fig. 8 based on target types. CPUs and MPUs showed high similarity, while CPUs and GPUs exhibited significant differences, making it challenging to generate accurate GPU code solely from CPU data. Additionally, VLIW and Virtual targets differed from mainstream CPUs due to variations in instruction sets, highlighting the need to use backend code from similar targets for training, as discussed in Sec. 4.3.1.

