# OpenReview forum: "ComBack: A Versatile Dataset for Enhancing Compiler Backend Development Efficiency"
_NeurIPS.cc/2024/Datasets_and_Benchmarks_Track — NeurIPS 2024 Track Datasets and Benchmarks Poster_

### Official Review · Reviewer_hMKy · 2024-07-26
**A Special-Domain Code Completion Dataset**

**Rating:** 7
**Confidence:** 4

**Review:**

The paper performed different training and test splits to ensure that training on their dataset would definitely help developers develop compilers for unseen hardware backends.
It showed an interesting observation that language models with only 220M parameters could have high accuracies on such tasks. Personally, I believe this could open the door to research that focuses on finetuning models on specific repos to help programmers update such repos.
I was personally concerned about leakage when first seeing the high accuracies of the finetuned models, but the authors have performed robust analysis on different training/test splits to ensure that the finetuned models could be used to develop compiler for unseen backends.

**Strengths:**

- Diversity in dataset: LLVM versions, hardware backends, compiler frameworks (GCC and LLVM)
- Robust analysis to ensure there is no data leakage
- Strong results after finetuning small language models

**Additional Feedback:**

- Throughout the paper there were no spaces between before parantheses. For example in the Abstract in line 14 "model(CodeT5+)" should be "model (CodeT5+)". This happened many times throughout the paper.
- Line 67: I think in papers, it is better to avoid abbreviations and write "need not"
- Line 91: When mentioning KLoC for the first time, please specify what it stands for
- Line 97: What does MPU stand for? Please specify it when mentioning the acronym for the first time
- Lines 136 to 139: Are you replacing all intermediate representations with the same token "<ISA_LIT>", all the enumeration variables with "<NUM_LIT>", etc.? If that's the case, aren't we losing information about the detailed instances of each category? Perhaps add a diagram for showing how a code sample looks like before and after replacement.
- Table 4 results are interesting. If the model was not finetuned on any code related to MPU and GPU, how was it able to generate immediate values, strings, and enums related to those backends?

**Clarity:**

Overall the paper was easy to read, but there are some comments:

- Table 3: Suggest to add a column specifying whether a model was finetuned or not
- Tables 3 and 4: Both tables doen't show what is the difference between the first 3 rows and the last 3 rows. There seems to be a missing column to distinguish between them.

**Correctness:**

- Line 235: "mainstream LLMs are still not well-suited for backend development tasks": Not sure if I agree with this statement. It's not surprising that mainstream LLMs underperform on special domain datasets and can improve if finetuned on them. I agree that it was surprising that finetuning small LLMs exceeded large LLMs significantly, but this claim could be re-worded to be less controversial.

**Documentation:**

- Line 238: it is not clear what is the Fork-Flow process here. Is it like individual human programmers were asked to implement the functions? Were these programmers experienced in those backends? Did they have access to a backend's documentation?

**Limitations:**

- Haven't evaluated more recent language models that have higher accuracy and larger context window. I understand though that is due to limited GPU resources. By open sourcing the dataset, other researchers could try on larger models.

**Opportunities For Improvement:**

- The selected models are old. Perhaps use recent models like Gemma2B, CodeGemma2B. If you can only try small models, you can try Apple's OpenELM 270M model.
To save memory space during training, you can use LoRA or QLoRA using frameworks like HuggingFace's TRL or PyTorch's torchtune.

- Models had very small context lengths (samples with input exceeding 256 tokens or ground truth surpassing 512 tokens were removed). If you consider finetuning with larger models as mentioned before, you may support more tokens. Even if available GPU resources won't enable training on those large samples, I recommend to keep them in the HuggingFace datasets for researchers to try finetuning on.

- Interested if you can evaluating the dataset on the recent LLM Compiler model: https://huggingface.co/collections/facebook/llm-compiler-667c5b05557fe99a9edd25cb. Although it was not trained specifically on code completion for compiler backend, but I am interested to know if it will perform better or worse than Code Llama given that it was trained on LLVM data. **I acknowledge though that this LLM Compiler model was only open sourced during the review period, so the authors are not obliged to try it.**

**Relation To Prior Work:**

I believe the Related Work section is concise. But I suggest to cite papers and tools for automatic compiler backend generators that do not necessarily use machine learning:

- Florian Brandner, Viktor Pavlu, and Andreas Krall. 2013. Automatic generation of compiler backends. Softw. Pract. Exper. 43, 2 (February 2013), 207–240. https://doi.org/10.1002/spe.2106

**Summary And Contributions:**

ComBack is a special-domain C++ dataset that focuses on developing compiler backends, particularly the LLVM and GCC codebases.
It consists of training and test splits.
The authors have shown that finetuning small sized LLMs on their dataset could lead to significantly high accuracy to help compiler backend developers speed up their coding.

Developing compiler backends for new versions of LLVM and GCC as well as for new versions of hardware is a tedious process that requires many software engineering hours. Finetuning LLMs on this dataset could help speedup the development process significantly.

---

> ### Author Rebuttal · Authors · 2024-08-17
>
> #### Q1. Experiment for a new language model (OpenELM 270M).
>
> We select Apple's OpenELM 270M for conducting the experiment in Section 4.2 in our paper, evaluating its accuracy improvement based on ComBack. We conducted a full fine-tuning on 8 Tesla V100 GPU for 30 epoch with ComBack, the results are listed in the attached pdf file.
>
> After being fine-tuned with ComBack, OpenELM-270M demonstrated a significant improvement in accuracy across all three tasks, proving that ComBack has also enhanced backend development abilities for the new language model as well.
>
> #### Q2. Data of longer context length.
>
> Thank you for your suggestion. Although our computational resources are insufficient to support longer contexts, we will make this data publicly available on Hugging Face for use by other researchers.
>
> #### Q3. LLM compiler model.
>
> Although our request to use the LLM Compiler on Hugging Face was declined by its developers, we will still discuss its potential applications in backend development by referencing their paper.
>
> The paper **"Meta Large Language Model Compiler: Foundation Models of Compiler Optimization"** details how the LLM Compiler, pre-trained and fine-tuned on Code-LLaMA using a dataset rich in LLVM IR, assembly, compiler emulation, flag tuning, and disassembly, excels in code optimization tasks. However, the limited backend-specific training data and the low accuracy of the original Code-LLaMA in this area (as evidenced in Section 4.3 in our paper) indicate room for improvement. We believe that fine-tuning the LLM Compiler on ComBack could enhance its backend development capabilities, advancing both compiler and AI research.
> #### Q4. Statement in Line 235.
>
> Thanks. We agree with your point and will rewrite the sentence to make it non-controversial in the revised version.
>
>
> #### Q5. Clarity for Table.3 and Table.4.
>
> We will add corresponding columns in Table.3 and Table.4. Thanks for your advice.
>
> #### Q6. Related Work
>
> Thanks. We will include this reference in the revision. It is indeed significant for automated backend generation.
>
> #### Q7. Description for the Fork-Flow process
>
> Fork-Flow refers to the traditional backend development process in which a developer first identifies a similar backend based on their experience. For example, when developing a backend for a CPU, the developer will look for an existing CPU backend, and when developing for a GPU, they will search for an existing GPU backend. After finding a similar backend, the developer modifies its functions to ensure correct operation on the new backend. This approach, compared to starting from scratch, reduces repetitive development effort to some extent.
>
> In the Fork-Flow experiment detailed in Section 4.3.1, we simulated the process using scripts rather than relying on programmers. We calculated the similarity (BLEU-4, Edit Distance) between the new backends (RISC-V, ARC, NVPTX) and existing backends of the same type (CPU, MPU, GPU), identifying the most similar match for each. As shown in Fig. 6, Fine-tuned CodeT5+ outperformed the maximum similarity achieved through Fork-Flow, demonstrating that even under optimal conditions, using Fine-tuned CodeT5+ for backend development is more efficient than traditional methods, further validating the value of ComBack.
>
> #### Q8. Spaces between before parantheses and abbreviations in the paper
>
> Thanks. We will thoroughly fix them in the revised version.
>
>
> #### Q9. Specifying the meaning of KLoC and MPU
>
> Thanks. We will add the specification for KLoC (Kilo Lines of Code) and MPU (Micro-Processor) in the revised version.
>
>
> #### Q10. Are you replacing all intermediate representations with the same token "<ISA_LIT>", all the enumeration variables with "<NUM_LIT>", etc.? If that's the case, aren't we losing information about the detailed instances of each category?
>
>
>
> * For two code completion tasks (statement-level completion and next-statement suggestion), we replaced all target-specific values with '<xxx_LIT>', etc. The reasons are as follows:
>
>    1. Target-specific values are determined by the hardware characteristics of the target and vary significantly across different targets, showing minimal regularity. Thus, similar to the removal of uncommon literals such as IP addresses and phone numbers in the CodeXGlue dataset, we replaced these values entirely to help the model focus more effectively on the remaining code.
>
>    2. Such target-specific values typically come from hardware documentation, making it challenging for the model to accurately predict these values based solely on the previous statement or part of the current statement. Therefore, we believe that a developer-friendly approach is to predict the rest of the code accurately and use a unified identifier to mark these target-specific values. This allows developers to quickly locate and insert the corresponding target-specific values.
>
> * For code generation tasks, we retained all target-specific values and related information. This is because such information is part of the user input and, additionally, in code generation data, the model can be trained on complete functions, enabling it to better predict the corresponding values.
>
> We will add a diagram for illustrating this in the revised version.
>
>
> #### Q11. Table 4 results are interesting. If the model was not finetuned on any code related to MPU and GPU, how was it able to generate immediate values, strings, and enums related to those backends?
>
> Based on answers on Q10, for two code completion tasks, as we replaced all target-specific values with '<xxx_LIT>', the model then can accurately predict some fixed code patterns for MPU and GPU. For code generation tasks, since the user input includes immediate values, strings, and enums, the model can generate the corresponding code.

---

> > ### Comment · Reviewer_hMKy · 2024-09-08
> >
> > Thanks for your detailed rebuttal addressing my comments. I would like to keep my score.

---

### Official Review · Reviewer_HjXi · 2024-08-02
**Great work in**

**Rating:** 7
**Confidence:** 4

**Review:**

Here's an evaluation of the paper's quality, clarity, originality, and significance, along with pros and cons:

**Quality**:
The paper demonstrates high-quality research with a well-structured methodology, comprehensive experiments, and thorough analysis. The authors have put considerable effort into creating a large-scale dataset and evaluating it across multiple models and tasks.

**Clarity**:
The paper is generally well-written and organized logically. The authors clearly explain the motivation, methodology, and results. However, some technical details might be challenging for readers unfamiliar with compiler development.

**Originality**:
The work is highly original, as it introduces the first public dataset specifically designed for compiler backend development using language models. This addresses a significant gap in the field.
Significance:
The work has substantial significance for both the compiler development and machine learning communities. It has the potential to significantly accelerate compiler backend development, which is crucial in the era of diverse and rapidly evolving processor architectures.

**Pros**:
* Addresses a critical need in compiler development with a novel approach.
* Large-scale, diverse dataset covering multiple compiler backends and target types.
* Comprehensive evaluation across multiple pre-trained language models.
* Demonstrates practical improvements over conventional methods and larger language models.
* Shows potential for generalization to new target types and extensibility for customized targets.
* Provides a valuable resource for future research in this area.

**Cons**:
* The paper doesn't discuss potential limitations of using language models for compiler backend development, such as ensuring correctness and handling edge cases.
* While the dataset is large, it's unclear how representative it is of the full range of compiler backends and target architectures.
The evaluation focuses on code completion and generation tasks, but doesn't address the overall quality or correctness of the generated code in a real compiler setting.
* The paper doesn't discuss the computational resources required for training and using these models in a practical development setting.
* There's limited discussion on how this approach might integrate with existing compiler development workflows and tools.
* The paper doesn't address potential privacy or licensing concerns related to using open-source code for training.

Overall, despite these limitations, the paper presents a significant contribution to the field, opening up new possibilities for applying machine learning to compiler development.

**Strengths:**

* Significance of the contribution:

Introduces ComBack, the first public dataset for compiler backend development using language models.
Addresses a critical gap in the field, as backend development is time-consuming and largely manual.
Demonstrates potential for significantly improving efficiency in compiler development.


* Relevance to the broader research community:

Bridges the gap between compiler development and machine learning communities.
Provides a valuable resource for researchers in both fields to build upon.
Opens up new research directions in applying AI to systems software development.

* Quality of the research:

Comprehensive dataset covering 178 backends from major compilers (GCC and LLVM).
Well-designed experiments evaluating multiple pre-trained language models.
Thorough analysis comparing fine-tuned models against conventional methods and larger language models.
Demonstrates generalizability to new target types and extensibility for customized targets.

* Ethical and social implications:

Potential to accelerate development of compilers for diverse hardware, supporting broader access to computing resources.
May lower barriers to entry for compiler development, fostering innovation.
Does not contain personally identifiable information or offensive content, mitigating privacy concerns.

* Practical impact:

Shows concrete improvements over conventional backend development methods.
Potential to reduce development time and costs for compiler backends.

* Extensibility:

Demonstrates ability to assist with new target types and support iterative expansion.
Suggests long-term usefulness as new hardware architectures emerge.

* Methodological robustness:

Evaluates performance across three distinct, relevant tasks.
Compares against both traditional methods and state-of-the-art language models.

* Clear presentation:

Well-structured paper with clear explanations of motivation, methodology, and results.
Provides detailed information about the dataset and experimental setup.

These strengths collectively present ComBack as a significant, high-quality contribution with broad relevance and positive implications for the field of compiler development and beyond.

**Additional Feedback:**

No additional feedback

**Clarity:**

The paper is generally well-written and organized logically. The authors clearly explain the motivation, methodology, and results.

**Correctness:**

Based on the information provided in the paper, the claims made in the submission appear to be largely correct and supported by the presented evidence. The dataset (ComBack) and the associated benchmark tasks are constructed in a sound manner.

**Documentation:**

The paper provides substantial detail on data collection, organization, and availability.

Data Collection and Organization:

The paper describes the data sources (GitHub repositories, official GCC and LLVM websites) and the versions covered (Section 3.2).
The data collection process is outlined, including crawling methods and filtering criteria.
The preprocessing steps are detailed, including comment removal, function extraction, and handling of target-specific values.

Availability and Maintenance:

The authors provide a URL for accessing the dataset (https://huggingface.co/datasets/docz-ict/ComBack).
They also provide a URL for accessing the fine-tuned models (https://huggingface.co/docz-ict/ComBack_Models).

**Ethics:**

No ethics concern

**Limitations:**

Addressed:
* The authors acknowledge in Section 6 that ComBack lacks function descriptions for highly-customized functions in backends for specific targets, identifying this as a limitation to be addressed in future iterations.
* They state that ComBack does not contain personally identifiable information or offensive content, mitigating potential negative societal impacts in this regard.
* The authors discuss some limitations of their approach, such as the decrease in accuracy when generalizing to new types of targets (Section 4.3.2).

However, there are several areas where the authors could more thoroughly address limitations and potential negative impacts:
* Integration challenges: The authors could address potential challenges in integrating this approach into existing compiler development workflows.
* Scalability and generalizability: More discussion on the limitations of the approach when dealing with radically new architectures or programming paradigms would be beneficial.

**Opportunities For Improvement:**

* Practical impact:

The computational resources required for training and using these models in practical settings are not discussed, which could be a barrier to adoption.
The paper doesn't explore how this approach might affect the skills required for compiler development, potentially raising concerns about deskilling in the field.

* Extensibility:

While the paper shows some ability to generalize to new target types, it's unclear how well this approach would scale to radically different architectures or programming paradigms.

* Methodological robustness:

The paper doesn't report error bars or discuss the variability of results across multiple runs.
There's limited ablation studies or exploration of how different components of the approach contribute to its performance.

**Relation To Prior Work:**

The paper does a reasonably good job of discussing how ComBack differs from previous contributions.

Novelty: The authors clearly state that ComBack is "the first public dataset designed for improving compiler backend development capabilities of language models" (Abstract and Section 1). This claim of novelty is a key differentiator.

Context in related work: In Section 5 (Related Work), the authors discuss existing work in backend development, datasets for compilers, and AI for compilation. They highlight that while there are datasets for other aspects of compilation (e.g., CodeXGlue, Compile, TenSet, ANGHABENCH), there is a "dearth of datasets tailored for compiler backends within the community."

Comparison with conventional methods: The paper compares ComBack's performance against the traditional Fork-Flow method, clearly showing how their approach differs from and improves upon conventional practices (Section 4.3).

Comparison with general-purpose language models: The authors compare their fine-tuned models against ChatGPT and Code-LLaMA, demonstrating the specific advantages of their approach in the compiler backend domain (Section 4.3).

**Summary And Contributions:**

This paper introduces ComBack, the first public dataset designed for improving compiler backend development capabilities of language models. The key contributions of this paper are:

**ComBack Dataset**: A large-scale dataset containing 178 backends for mainstream compilers (GCC and LLVM), covering various target types like CPUs, GPUs, etc. It includes over 180,000 functions and 5.7 million lines of code.
Three Tasks: The dataset supports three tasks representative of common backend development scenarios:

* Statement-Level Completion
* Next-Statement Suggestion
* Code Generation

**Effectiveness Evaluation**: Experiments show that fine-tuning six pre-trained language models on ComBack significantly improves their performance across the three tasks.
Efficiency Improvement: The fine-tuned CodeT5+ model (220M parameters) outperformed conventional backend development methods (Fork-Flow) as well as larger models like ChatGPT-3.5-Turbo and Code-LLaMA-34B-Instruct on the three tasks for new targets.
Extensibility: ComBack demonstrated the ability to assist in development for new types of targets and support iterative expansion for customized targets.

The authors argue that ComBack can potentially improve the efficiency of compiler backend development by enabling language models to better assist programmers in code completion and generation tasks. This is significant given the manual effort currently required in backend development for diverse processor targets.

---

> ### Author Rebuttal · Authors · 2024-08-17
>
> #### Q1. Correctness and edge cases of using language models for compiler backend development.
>
> When using language models for backend development, errors in code completion or generation are inevitable. However, the real value lies in offering code suggestions to reduce the programmer's workload, as discussed in lines 147-151 of our paper. While this approach aids development, the generated code must be verified or modified by the programmer. Our goal is not to fully automate code generation, but to assist developers while they ensure the correctness.
>
> #### Q2. How representative it is of the full range of compiler backends and target architectures.
>
> As detailed in Section 3.1 and Table 1, ComBack includes 178 targets for GCC and LLVM, covering a wide range of processor types such as CPU, MPU (Microprocessor), GPU, VLIW, DSP, and Virtual ISA. Additionally, Table 6 in Appendix A provides a detailed list of the 178 targets, showing that it includes the most widely used and representative targets across different categories, such as X86, ARM, RISC-V, and MIPS in CPUs; ARC and AVR in MPUs; AMDGPU and NVPTX (Nvidia GPU) in GPUs; and Hexagon in DSPs, among others. In summary, the target architectures in ComBack are highly representative.
>
> #### Q3. The overall quality or correctness of the generated code in a real compiler setting.
>
> For the two code completion tasks (statement-level completion and next-statement suggestion), we used Exact Match as the evaluation metric, requiring the generated code to be identical to the ground truth, with accuracy detailed in Table 2 - Table 5. In code generation, while most model-generated functions do not achieve exact matches or full correctness (i.e., compiling and running tests correctly), they still require user modifications. However, as shown in Fig. 6, the model-generated code is more similar to the correct implementation than with the traditional Fork-Flow method, reducing the user's modification burden.
>
> #### Q4. The computational resources required for training and using these models in a practical development setting.
>
> In line 203, we mention that the experimental platform used in this paper is equipped with 8 NVIDIA Tesla V100 GPUs, each with 16GB of memory. Training for approximately 30 epochs typically requires 48-72 hours. However, when using the fine-tuned model, it can be deployed on a single V100 with 16GB of memory, achieving rapid code completion and generation.
>
>
> #### Q5. Discussion on how this approach might integrate with existing compiler development workflows and tools.
>
> As discussed in lines 147-151 of our paper, this approach enables automatic code completion based on the developer's real-time context and generates entire functions from natural language descriptions, similar to using Copilot which can be integrated into VSCode to assist developers. Even within traditional development processes like Fork-Flow, it can enhance efficiency through automatic code completion and function generation, making it a valuable addition to existing backend development methods.
>
> #### Q6. Potential privacy or licensing concerns related to using open-source code for training.
>
> * **Privacy**: The data in ComBack is sourced entirely from open-source compiler code and contains no personal information (as discussed in lines 297-298), so it does not pose any potential privacy concerns.
>
> * **Licensing Concern**: ComBack's data comes from open-source code on GitHub and official releases. All data and models are available on Hugging Face and are not used commercially. Thus, under common licenses like MIT and BSD, our dataset does not face licensing issues. Besides, we have also noted the emergence of non-AI licenses on GitHub (https://github.com/non-ai-licenses). We have re-examined the repositories involved in our dataset and confirmed that none of them use non-AI licenses, ensuring there are no licensing concerns.
>
> #### Q7. Discussion on how well this approach would scale to radically different architectures or programming paradigms.
>
> * **Different Architectures**: In Section 4.3.2, we conduct an experiment where the CPU serves as the training/validation set, while the MPU and GPU are used as the test set to evaluate ComBack's generalizability across different architectures. The results presented in Table 4 demonstrate that ComBack can effectively facilitate code completion and generation for different architectures, though there is a noted decrease in accuracy when compared to training and testing on the same architecture.
>
> * **Different Programming Paradigms**: Currently, ComBack is primarily tailored for two mainstream compilers: GCC and LLVM. Since backend compiler development must adhere to the function interfaces and programming conventions provided by the compiler infrastructure, ComBack does not yet offer robust support for other programming paradigms, such as a fully independent compiler designed by a developer. However, we plan to iteratively expand ComBack to more effectively support various programming paradigms in the future.
>
> #### Q8. Discussion on the variability of results across multiple runs and potential ablation studies.
>
> * **The variability of results across multiple runs**: Currently, since we have fixed all model parameters (including the random seed), the results on the test set show no variability. In future work, we plan to explore different parameter combinations to investigate variations in the model's output.
>
> * **Ablation studies**: In our experiments, we employed a basic sequence-to-sequence training approach without additional optimization techniques. Moving forward, we will design further optimization strategies to enhance the accuracy of the model in code completion and generation tasks, and conduct ablation studies to assess their impact.

---

### Official Review · Reviewer_3S26 · 2024-08-16
**A Dataset and Mini-Benchmark for Finetuning LLMs for Compiler Backend Development**

**Rating:** 7
**Confidence:** 4

**Review:**

Originality: The authors identify the lack of a code dataset for fine-tuning LLMs on compiler backends. They proceed the contribute the first instance of a dataset specifically curated for this purpose. They also provide a mini-benchmark to evaluate LLMs fine-tuned on their dataset for compiler backend code generation. I could not find a precedent for such a suite of evaluation tasks in the literature. Therefore, I deem this an original piece of research, albeit it targets quite a narrow domain of data.

Quality: The research problems that motivate the paper are clear and worth investigating: Are the mainstream LLMs for code generation adequate for generating compiler backend code? Could a specialized dataset curated for this type of codebase improve existing LLMs' accuracy on this domain of code generation? Preliminary experiments conducted on a sufficiently large and diverse set of LLMs answer the first question clearly: there is room for improvement. The dataset they curated and subsequent fine-tuning experiments improve the baselines across all three tasks. The data collection and pre-processing steps are transparently described, most design decisions are clearly explained, and the paper maintains a technically-rich yet easily understandable narrative. Therefore, I find this to be a high quality work.

Clarity: The authors strike a good balance in providing just enough amount of detail when introducing the background of the work, namely, the anatomy of a compiler, software compilation process, and the challenges surrounding compiler backend development. Their course of action to formulate research problems and attack them are easy to follow. Their design decisions on various steps of this project are transparently communicated. Even though I am not an expert on compilers, as a deep learning researcher specializing on LLMs, I was able to easily follow the paper from beginning to end. It is a well-written paper.

Significance of the Work: This paper targets a very narrow and specific domain of data, namely, compiler backend source code. However,  since most software is written in high-level languages, and all high-level languages have to be translated to low-level executable machine code, streamlining the process of writing compiler backends that perform this task is a significant contribution to the broader software development community. Therefore, I believe this paper makes a significant enough effort to be featured in the proceedings of this conference.

Pros:
* The authors provide the first dataset for compiler backend source code and publish the dataset.
* The authors contribute a suite of evaluation tasks for an LLM to be evaluated on for compiler backend code generation.
* The paper is written clearly, with transparency around technical details and design decisions.
* Research questions are clearly enumerated in the beginning of the experiments section, and each of them are answered by a dedicated experiment.

Cons:
* There are potential concerns about the way data is split into training, validation, and test sets. Current strategy of random sampling is agnostic to the distribution of inherited and custom functions in the data as well LLVM and GCC compilers. This might be skewing the results in some of the evaluations (see Opportunities for Improvement).
* When testing for the code generation for new backends, the authors seem to diverge from their strategy on the other tasks and keep the concrete values for the target-specific variable values. Models trained in this fashion might memorize and overfit to particular architectures and suggest incompatible code to developers. If accepted by the developer, this might lead to bugs that are hard to identify (see Opportunities for Improvement for further detail).
* Most design decisions are elucidated in the paper, however there are a few choices of significance that demand a clarification. Please see them listed in the Opportunities for Improvement section).

**Strengths:**

Most of the code written today is written in high-level languages, which need to be translated to executable machine code to run on hardware. Since compiler backends are an integral component in this process, addressing inefficiencies in their development is a worthwhile pursuit. As we observe this prolific era in the development novel hardware architectures, it is reasonable to employ powerful tools like large language models to streamline the process of developing compiler backends. Therefore, I find the contribution made in this work meaningful and timely.

The nature of the work presented here occupies a niche segment within the broad field of machine learning research. However, its potential implications for the development of software compilation infrastructure are relevant not only to the audience of this conference but also to the broader software development community.

Aside from a few suggestions I will make in the following sections, I am generally content with the quality of the research presented here. The authors point out to the inadequacy of mainstream LLM-based code generation methods in producing accurate code completion suggestions for compiler backends. They adapt three reasonable code generation tasks on which they test numerous LLMs to validate this observation. They then proceed to make a clear hypothesis, proposing that the root cause of this phenomenon is the lack of sufficient LLM training data for compiler backend code. They test this hypothesis by compiling a sizable open-source code dataset that targets hundreds of popular compiler backends. Finally, they prove the merit of such a dataset, which they claim did not exist before this work, by demonstrating an improvement on all baselines as a result of fine-tuning on their dataset.

I was satisfied by the clarity with which this paper was written. In particular, I appreciated the transparency around the data collection and pre-processing steps. The origin of the data was communicated clearly and the roadmap for compiling raw source code into a useful LLM fine-tuning dataset was outlined. Additionally, the authors explain their choices for the model choices, design decisions for the prompt templates, the research question investigated at each experiment conducted, their training settings, and various hyperparameter configurations.

While I was reading the early stages of the paper, I was concerned that the dataset would contain hard-coded values for the target-specific variables, potentially resulting in the fine-tuned LLMs overfitting to particular architectures. This would beat the purpose of the dataset, which is the to train LLMs that can generalize to rapidly incoming novel compiler backends. However, the authors addressed this concern in section 3.2.4 by replacing these values with intermediate representations. They also design a set of experiments (section 4.3.1 and 4.3.2) to ensure that the models can generalize to unseen architectures within and across device types.

**Additional Feedback:**

Thank you for your submission to the NeurIPS 2024 Datasets and Benchmarks Track. It was a pleasure to review your work. Best of luck!

**Clarity:**

The authors strike a good balance in introducing adequate detail about the anatomy of a compiler so as to make their contributions and their research problems clear to a machine learning researcher who is not familiar with the specifics of compiler backend development.

I commend the authors for clearly enumerating their research questions in the beginning of the experiments section and clarifying which of those questions are answered by each experiment. This bolsters the academic rigor of the paper and makes it easier to read.

The authors adopt a transparent approach throughout the paper regarding decisions and choices they made at each steps. Even though I have noticed a few exceptions to this commitment (which I pointed out in Opportunities for Improvement), I appreciated observing an effort towards transparency and facilitating reproducibility.

**Correctness:**

The authors claim that they are contributing the first publicly available dataset for compiler backend source code. My literature review has not yielded a precedent to debunk their claim. They follow a reasonable and transparent roadmap for compiling their dataset. I found their curation method conscientious about the software licenses and copyrights.

Although the highlighted contribution is the dataset, the authors also provide a mini-benchmark of three evaluation tasks in this paper. I find their chosen tasks appropriate for and aligned with the real-life software development tasks. Even though I have minor concerns about their specific sampling strategies, I find their experiments rigorous. In particular, I appreciate the fact that they evaluate the generalization capabilities to cater towards new architectures separately. Therefore, I approve of the overall methodology and experiment design of the work presented here.

**Documentation:**

All the datasets, codes, and instructions for reproducing each table in the manuscript are provided on the Huggingface platform. A URL to the repository is provided in the manuscript.

**Ethics:**

I do not suspect an ethical concern since the code is sourced from public Github repositories and official LLVM and GCC websites. Licenses for baseline models are provided in the appendix and for the purposes of this research, the authors do not seem to be in violation of those licenses.

**Limitations:**

The authors first describe a masking strategy to replace concrete values of target-specific variables with uniform masking tokens. However, for evaluating code generation for new backends of existing and new types, they revert this decision and use the concrete values. I am concerned that this might lead to inherently probabilistic LLMs to memorize or hallucinate those values when suggesting code completions. If accepted as-is by the developer, this might lead to seemingly correct but actually target-incompatible code that will be quite challenging to debug. I suggest the authors revise their loss computation during fine-tuning, and score computation during evaluation to address the issue of potential mode collapse, but still keep the mask tokens. A model that suggests a correct template with masks for concrete values will be a lot more useful to the developer.

I also urge the authors to revise their train-validation-test data split strategy. I think random sampling is content agnostic and superficial given the reportedly imbalanced distribution of data they have compiled. This imbalance in noticeable in the compiler (LLVM vs GCC) and function-type (inherited vs custom) axes.

**Opportunities For Improvement:**

1. As you pointed out, inherited functions share a template and most of the time target-specific values are the only changes required to cater towards a backend. When you randomly split the data, you might leak some of those common patterns into the test data, inflating the results. This might explain why the improvements are so dramatic in Table 2. I HIGHLY recommend you evaluate inherited and custom functions separately.

2. In section 4.2 (Accuracy Improvement Across Various Models), authors report that they split the backend data into randomly into training, validation and test sets. I believe the sampling strategy employed here might call for a more content-aware approach. It is reported that ComBack features 4,847.5 KLoC of data from LLVM backends and 883.7 KLoC of data from GCC backends, a clearly imbalanced distribution in favor of LLVM backends. This might be a confounding factor in the results reported in Table 3, where we observe CodeT5+ (fine-tuned on ComBack) perform better on all LLVM backends with respect to GCC backends (except for one). Notice that ChatGPT-3.5-Turbo and Code-LLaMA baselines are fairly consistent across GCC and LLVM backends, because they were not fine-tuned with this randomly sampled data.

3. I am familiar with the metrics chosen in Section 4.1, however, I would recommend adding a one-sentence description of them and explaining why they were chosen for an audience who may not be familiar with them. Even as much as clarifying whether a higher or lower score is desired would be helpful.

4.  I recommend explaining your choice of ChatGPT-3.5-Turbo and Code-LLaMA-34B as baselines for the code generation task. This does not mean I do not approve of them. Given that OpenAI Codex evolved into ChatGPT-3.5-Turbo and Code-LLaMA-34B performs well on HumanEval, I believe you set a reasonable bar for the baselines. I would like to see a brief discussion making this clear.

5. In section 3.3.3, the authors report that they replace the intermediate representations with the corresponding target-specific value in both the input and ground truth data. I am not in favor of this decision as it will lead to a model that will suggest code with concrete target-specific values. If the developer accepts the suggestions and forgets to replace the target-specific values with those of the new backend he's writing, it will lead to bugs downstream, and will be quite hard to pinpoint the wrong value to debug it. I was very pleased to see the strategy you implemented in section 3.2 for the other two tasks, I suggest you do the same here as well. It makes the fine-tuned models backend-agnostic. I understand that this might inflate the accuracy results on the evaluation task if the model falls into mode collapse on the repeated intermediate representation tokens, but you can simply choose to not score those tokens during fine-tuning and evaluation.

6. In table 3, the alignment is somewhat off. I recommend adding separator columns between EM and ED values for each backend to improve readability of the results.

6. In table 4, I think the anomaly observed with NVPTX (GPU) deserves to be explained. What do you think is the reason for slight decrease in accuracy even though the model sees the GPU and MPU backends? Perhaps you can at least speculate why this might be.

7. I was happy to see that you paid attention to potential leakage into test data due to the similarity between RISC-V (CPU) and RI5CY in LLVM. Have you considered the same risk for ARC and NVPTX, as well?

**Relation To Prior Work:**

The authors provide a brief discussion organized into three subsections: backend development, datasets for compilers, and AI for compilation. I find these three angles to be appropriate principal components for a related work discussion for this paper. That said, I would expect a finer granularity in the way the previous work is discussed. For instance, the authors mention that Processor Design Languages for proposed to mitigate manual efforts "to some degree". I would be more rigorous of them to communicate how those seven approaches they cited differ from each other, and more importantly, from their own approach.

**Summary And Contributions:**

The authors highlight the challenges posed by the rapid development of novel hardware architectures for compiler backend developers, who must adapt to these new architectures through manual, time-consuming, and often redundant programming efforts. They recognize the potential of Large Language Models (LLMs) for code generation and propose that these models could significantly boost developer efficiency in compiler backend development. The paper motivates its work by asserting and experimentally validating that prominent mainstream LLMs, including those fine-tuned for code generation, remain inadequate for the specific sub-domain of compiler backend development. This inadequacy is attributed to the lack of a targeted code dataset, which the authors address with the introduction of ComBack, 'a versatile dataset for enhancing compiler backend development efficiency.'

The authors contribute the first public dataset designed for fine-tuning LLMs for compiler backend development, which they make available on the Huggingface platform. Named ComBack, the dataset is compiled from 317 public GitHub repositories and official source code from the GCC and LLVM websites. It includes source code for 178 mainstream compiler backends, with 77 targeting GCC and 101 targeting LLVM compilers. ComBack also features a suite of three downstream evaluation tasks to assess LLM performance in statement-level completion, next-statement suggestion, and code generation. To evaluate the utility of their dataset, the authors assess six pre-trained LLMs on these tasks and compare the performance of these models after fine-tuning on ComBack. Their evaluation demonstrates that ComBack improves model accuracy across all models and tasks. They further evaluate the top-performing fine-tuned model (CodeT5+) on its ability to generalize to new backend targets not included in the fine-tuning set, using pre-existing LLMs (ChatGPT-3.5-Turbo and Code-LLaMA-34B-Instruct) and conventional methods (Fork-Flow) as baselines. They report that CodeT5+ outperforms baseline methods in terms of exact match accuracy (EM), Levenhstein Edit Distance Similarity (ED) and BLEU-4 metrics, despite its relatively small parameter count of 220 million.

---

> ### Author Rebuttal · Authors · 2024-08-17
>
> #### Q1. Data leakage in inherited functions.
>
> Yes, inherited functions usually have more fixed patterns, resulting in better test set performance after fine-tuning. In contrast, customized functions lack regularity, leading to poorer test results even with fine-tuning. Due to time constraints, we will later evaluate the data in Table 2 by separating it into inherited and customized functions.
>
> #### Q2. The sampling strategy employed might call for a more content-aware approach.
>
> LLVM's superior modular design makes it more widely developed and customized than GCC in academia and industry, which is why our dataset contains more LLVM samples. Consequently, the fine-tuned CodeT5+ performs better on LLVM-related tasks. We will conduct an experiment with equal amounts of GCC and LLVM data, further fine-tune the model, and compare the results with our current findings.
>
> #### Q3. Explanation for choice of evaluation metrics.
>
> We will explain our choice of evaluation metrics in the revised version. Exact Match was used for the two code completion tasks because it directly measures the correctness of the generated code, meeting developers' needs in real-time programming. For Code Generation, we chose BLEU-4 to assess structural similarity between the generated code and the ground truth, the higher the BLEU-4 score, the greater the similarity. We also used edit distance for all tasks to measure the modifications needed to align the generated code with the ground truth, where a higher score indicates fewer required edits and closer alignment to the ground truth.
>
> #### Q4. Explanation for choice of ChatGPT-3.5-Turbo and Code-LLaMA.
>
> In our revised paper, we will explain our choice of ChatGPT and Code-LLaMA. ChatGPT is the most widely used LLM globally, while Code-LLaMA, an open-source LLM designed specifically for code-related tasks, achieves state-of-the-art performance on many code related benchmarks.
>
>
>
> #### Q5. Replacement of intermediate representations in the Code Generation task.
>
> We appreciate your suggestion. Keeping intermediate representations unreplaced helps the fine-tuned model become backend-agnostic and improves accuracy in the remaining code. Therefore, we will add a code generation dataset with unreplaced intermediate representations to our HuggingFace repository, allowing other researchers to fine-tune backend-agnostic models. Thank you for your recommendation.
>
>
>
> #### Q6. Adding separator columns between EM and ED values for each backend to improve readability of the results in Table 3.
>
> We will add separator columns in the revised version. Thanks.
>
>
>
>
> #### Q7. The reason for slight decrease in accuracy even though the model sees the GPU and MPU backends? Perhaps you can at least speculate why this might be.
>
> Our analysis identifies two key reasons:
>
> 1) Table 1 shows that our dataset includes only two GCC GPUs: NVPTX and AMDGPU. Figure 6(a) indicates that AMDGPU and NVPTX have low similarity (edit distance 3.81, BLEU-4 4.70). Thus, the GPU data for GCC in the train set (AMDGPU only) is inherently less similar to NVPTX in GCC.
>
> 2) Inherited functions often share similar patterns, allowing the model to learn patterns from CPU data that are also relevant to some NVPTX functions.
>
> Therefore, the low similarity between GCC GPU data (AMDGPU only) and NVPTX, along with the model's ability to learn from CPU data, results in improved accuracy on NVPTX despite removing GPU data from the train set.
>
> #### Q8. Potential of data leakage of NVPTX and ARC.
>
> Yes, we considered this issue. Unlike RISC-V, which is open-source and supports various customized designs and extensions such as RI5CY, NVPTX and ARC do not have additional variants in our dataset.
>
> #### Q9. Clarity for research questions and experiments.
>
> Thanks. We will clarify the relationship between three research questions and experiments in Section 4 in our revised paper.

---

### Official Review · Reviewer_MbzQ · 2024-08-16
**Good work on extending LLM capabilities to backend compiler development; clear accept with minimal revision.**

**Rating:** 8
**Confidence:** 3
**Correctness:** The claims in the paper are correct.

**Review:**

This study contributes a new dataset of compiler backend source code in order to train LLMs to assist in code generation for diverse target backends. In addition to providing a dataset, the authors develop three tasks to measure the quality of LLMs in backend compiler development. To validate the utility of their dataset, the authors run several LLM fine-tuning experiments to evaluate the generalizability of their approach and compare it to existing state-of-the-art methods for code generation. Due to the clarity with which this work is presented and the empirical evidence provided in the results, I recommend this work for acceptance. I’ve noted a few minor corrections that should be made including; (1) uploading crawling code to aid in reproducibility, (2) including more descriptive data statistics, and (3) clarification on how fine-tuning was conducted. In summary, this work should clearly be accepted as it adds a meaningful contribution to the open-source community by bringing code generation capabilities to backend compiler development and providing an initial benchmark for continued improvement.

**Strengths:**

1. Introduces a new dataset and set of benchmarks to promote the development of LLM-based automation in backend compiler development.

2. The paper is clear for readers without extensive background in compilers. They clearly describe the problem, scope of research/application, and motivation for experiments.

3. The results clearly explain how the datasets and benchmark tasks were constructed, provides clear evidence that the dataset enhances the ability of fine-tuned LLMs over current state-of-the-art, and provides tests to evaluate the generalizability and overall feasibility of using the LLM-based completion for new target backends.

**Additional Feedback:**

N/A

**Clarity:**

The paper is very well written. A few minor points that would improve the clarity:
1. Please explicitly list the three tasks in the abstract/conclusion.
2. On line 100 change “model training” to “deep learning workloads” to encompass both training and inference.
3. On line 119 change “by” to “using”.
4. On lines 128-129 put the figure text references in order of 4(a), 4(b), 4(c).

**Documentation:**

The paper provides sufficient details and the data is well organized, documented, licensed, and downloadable using HuggingFace.

**Ethics:**

No ethical concerns.

**Limitations:**

The authors discuss limitations of the work including the absence of function descriptions for highly-customized functions in backends for specific targets for which they plan to address in future works. To the best of my knowledge, there are no potential negative societal impacts of the work.

**Opportunities For Improvement:**

1. Source code to reproduce the construction of the dataset and tasks should be provided to reproduce the contributions described in sections (3.2 Data Collection and Pre-processing) and (3.3 Tasks in ComBack). Uploading the crawling, pre-processing, data curation, etc scripts to the HuggingFace repository would be of benefit to the community.

2. In section (3.3.3 Code Generation) it would be helpful to include the percentage of functions that have natural language descriptions and comment on any bias in the types of functions with/without natural language descriptions.

3. In section (4.4 Iterative Expansion Ability), the authors should clarify whether they continued fine-tuning or if they restart fine-tuning from scratch using the dataset which includes RISC-V data. In specific, I’d just like clarification within this sentence from the paper: “then we add RISC-V into train and validation set and fine-tuned CodeT5+ with the new data.”

**Relation To Prior Work:**

Prior work is clearly discussed and referenced.

**Summary And Contributions:**

The study presents a dataset of compiler backend source code to aid the development of large language model (LLM) code completion methods. The dataset is built by crawling GitHub, the GCC website, and the LLVM website and then applying deduplication and additional crawling for associated function descriptions resulting in 883.7 KLoC for GCC and 4,847.5 KLoC for LLVM across 178 diverse target backends. They introduce three benchmark tasks for backend compiler code including (1) statement-level completion, (2) next-statement suggestion, and (3) code generation and conduct LLM fine-tuning experiments across the three tasks demonstrating state-of-the-art performance compared to open and closed source LLMs and conventional development methods.

---

> ### Author Rebuttal · Authors · 2024-08-17
>
> #### Q1. Source code to reproduce the construction of the dataset and tasks.
>
> Thanks for your suggestion, we will release source code for reproducing the construction of the dataset and tasks in our HuggingFace repository.
>
> #### Q2. Including the percentage of functions that have natural language descriptions and comment on any bias in the types of functions with/without natural language descriptions.
>
> Thanks. We will clarify in Section 3.3.3 that 68.08% functions in LLVM and 48.12% functions in GCC have natural language descriptions.
>
>
>
> #### Q3. Clarification for whether they continued fine-tuning or if they restart fine-tuning from scratch using the dataset which includes RISC-V data.
>
> We will clarify that we restart fine-tuning from scratch using the dataset which includes RISC-V data in our revised paper. Thanks.
>
> #### Q4. Clarity for writing issues.
>
> * List all three tasks in the abstract.
> * On line 100 change "model training" to "deep learning workloads" to encompass both training and inference.
> * On line 119 change "by" to "using".
> * On lines 128-129 put the figure text references in order of 4(a), 4(b), 4(c).
>
> Thanks for your constructive comments, we will fix all these issues in our revised version.

---

### Author Rebuttal · Authors · 2024-08-17

We thank all reviewers for your constructive comments.

## General Response

#### Q1. Corectness of using language models for compiler backend development.


As an AI approach, errors in using language models for backend development are inevitable. However, the true value of this approach lies in providing code completion and generation that reduces the programmer's workload. While this method supports the development process, the responsibility for verifying and adjusting the final code still rests with the programmer. The aim is not to fully automate code generation but to assist developers with improving development efficiency of compiler backends.


#### Q2. Integration of language models with existing compiler development workflows.

As detailed in lines 147-151 of our paper, language models fine-tuned with ComBack facilitates automatic code completion by analyzing the developer's real-time context and generating entire functions from natural language descriptions, akin to Copilot's functionality within VSCode. Even in traditional development workflows like Fork-Flow, it can significantly boost efficiency by automating code completion and function generation, making it a valuable enhancement to existing backend development practices.


#### Q3. Selection of language models in experiment.

We appreciate Reviewer hMKy's suggestion, and in response, we conducted additional experiments using Apple's OpenELM-270M. The results indicate that after fine-tuning with ComBack, OpenELM-270M achieved a significant accuracy improvement of all three tasks. Detailed results can be found in the attached pdf file of the author response.

#### Q4. Replacing Target-Specific Values in Code Completion and Code Generation.

Target-specific values vary widely across different targets due to hardware characteristics, showing minimal regularity. In statement-level completion and next-statement suggestion tasks, predicting these values accurately based on previous or partial statements is challenging. To help the model focus on code patterns, we replaced these values entirely for these tasks. However, for code generation tasks, we retained all target-specific values, as they are part of the user input.

We appreciate Reviewer 3S26's suggestion to replace all target-specific values in the code generation tasks, as it could be challenging for developers to identify incorrect values and debug effectively. In response, we will add an alternative version to our Huggingface Repository that includes a code generation dataset with the target-specific values replaced.



**As commented by all reviewers, the primary contribution of this paper is the introduction of the first open-source large-scale dataset specifically designed for backend development. Our experiments demonstrate that leveraging this dataset can significantly enhance the performance of language models in backend development tasks, such as code completion and generation, surpassing the accuracy of traditional development methods. We believe this dataset will attract more attention from researchers in the compiler and AI communities, further advancing the efficiency of backend compiler development.**

---

### Decision · Program_Chairs · 2024-09-26

**Decision:**

Accept (Poster)

**Comment:**

Thank you for your submission to the NeurIPS 2024 Datasets and Benchmarks Track. After careful consideration of the reviews and author responses, I am pleased to recommend this paper for acceptance. Here is a summary of the key points:

**Summary and Contributions **
The paper introduces ComBack, a public dataset specifically designed for improving compiler backend development capabilities of language models. Key contributions include:
* A large-scale dataset containing 178 backends for GCC and LLVM compilers, covering various HW target types.
* Three benchmark tasks representative of common backend development scenarios.
* Comprehensive evaluation showing significant performance improvements when fine-tuning pre-trained language models on ComBack.
4. Demonstrate that a smaller model (CodeT5+ with 220M parameters) when fine-tuned can outperform larger models and conventional methods on new targets.

**pros**
* Addresses a critical need in compiler development.
* Large-scale, diverse dataset covering multiple compiler backends and target types.
* Comprehensive evaluation across multiple pre-trained language models.
* Demonstrates practical improvements over conventional methods and LLMs.
* Shows potential for generalization to new target types and extensibility for customized targets.
* Clear presentation and thorough analysis of results.

**Areas for Improvement**
* Consider evaluating more recent LLMs with larger context windows.
* Expand discussion on integration with existing compiler development workflows.
* Expand on limitations and potential negative impacts in the discussion section.
* Evaluate the ability to generalize to new programming paradigms / HW targets.

Despite these areas for improvement, this work makes a decent contribution to the field of compiler backend development and opens up new possibilities for applying ML in this domain. The authors have addressed many of the reviewers' concerns in their rebuttal and have committed to incorporating the suggested improvements in the final version.

I recommend accepting this paper for publication at NeurIPS 2024 Datasets and Benchmarks Track.